# Large Language Models Engineer Too Many Simple Features for Tabular Data

## Abstract

Tabular machine learning problems often require time-consuming and labor-intensive feature engineering. Recent efforts have focused on using large language models (LLMs) to capitalize on their potential domain knowledge. At the same time, researchers have observed ethically concerning negative biases in other LLM-related use cases, such as text generation. These developments motivated us to investigate whether LLMs exhibit a bias that negatively impacts the performance of feature engineering. While not ethically concerning, such a bias could hinder practitioners from fully utilizing LLMs for automated data science. Therefore, we propose a method to detect potential biases by detecting anomalies in the frequency of operators (e.g., adding two features) suggested by LLMs when engineering new features. Our experiments evaluate the bias of four LLMs, two big frontier and two small open-source models, across 27 tabular datasets. Our results indicate that LLMs are biased toward simple operators, such as addition, and can fail to utilize more complex operators, such as grouping followed by aggregations. Furthermore, the bias can negatively impact the predictive performance when using LLM-generated features. Our results call for mitigating bias when using LLMs for feature engineering.

## 1 Introduction

Machine learning problems for tabular data exist in many domains, such as medical diagnosis, cybersecurity, and fraud detection (Borisov et al., 2022a; van Breugel & van der Schaar, 2024). The original data for these problems (e.g., an Excel sheet) often requires manual feature engineering by a domain expert to solve the machine learning problem accurately (Tschalzev et al., 2024). During (automated) feature engineering, various operators (e.g., `Add`, `Divide`, `GroupByThenMean`) are applied to existing features to create new features (Kanter & Veeramachaneni, 2015; Prado & Digiampietri, 2020; Mumuni & Mumuni, 2024).

Large language models (LLMs) understand various domains (Kaddour et al., 2023; Kasneci et al., 2023; Hadi et al., 2024), tabular data (Ruan et al., 2024; Fang et al., 2024), and feature engineering (Hollmann et al., 2024; Jeong et al., 2024; Malberg et al., 2024). Thus, data scientists have started to leverage LLMs for feature engineering, especially via the use of CAAFE (Hollmann et al., 2024), a powerful method for automatic feature engineering with LLMs[1]; liberating practitioners from extensive manual labor.

Despite their utility, LLMs are known to have negative biases as observed for chat applications (Kotek et al., 2023; Navigli et al., 2023; Gallegos et al., 2024; Bang et al., 2024) or when "meticulously delving" into text generation (Liang et al., 2024). Observing such biases motivated us to determine whether LLMs also exhibit a bias that negatively impacts the quality of their engineered features. If a bias is found and can be circumvented, LLMs would become a more potent tool for data scientists.

To determine whether a bias exists, we inspect the frequency of operators LLMs use when engineering features. This parallels inspecting the frequency of words to detect LLM-generated text (Liang et al., 2024). Assuming LLMs use their world knowledge and reasoning capabilities to employ the most appropriate operator, we expect the operators' frequencies to be similar to those of the optimal operators. That is, if adding two features is often the optimal feature, then an LLM would frequently

---

[1]For example, see these recent Kaggle competition write-ups (Hatch, 2024; Türkmen, 2024).

add two features. Moreover, if the LLM does not know the optimal operator, it should resort to a random search over operators.

Therefore, we compare the frequencies of operators used by an LLM to those obtained by searching for the optimal features with automatic black-box feature engineering using OpenFE (Zhang et al., 2023). Using this approach, we evaluated the bias of 4 LLMs, namely GPT-4o-mini (OpenAI, 2024), Gemini-1.5-flash (Gemini-Team, 2024), Llama3.1-8B (Touvron et al., 2023), and Mistral7B-v0.3 (Jiang et al., 2023). We obtained the distribution over operators for 27 tabular classification datasets unknown to all LLMs.

Our results demonstrate that LLMs can have a negative bias when engineering features for tabular data. LLMs favor simple operators during feature engineering (e.g., `Add`), while some LLMs rarely use more complex operators (e.g., `GroupByThenMean`). In contrast, automatic black-box feature engineering favors complex operators but also uses simple operators.
In particular, we observed a strong bias and negative impact for GPT-4o-mini and Gemini-1.5-flash, two big frontier models (Chiang et al., 2024). Both select simple operators most often and their generated features decrease the average predictive performance. In contrast, Llama3.1-8B and Mistral7B-v0.3, the small open-source models, are less biased or negatively impacted. Nevertheless, no LLM is close to the distribution over operators obtained by OpenFE. Likewise, the features generated by OpenFE improve the predictive performance on average the most.

**Our Contributions.** Our long-term goal is to enhance LLMs for automated data science. This work contributes toward our goal by: **(1)** developing a method to analyze LLMs for bias in feature engineering, and **(2)** demonstrate the existence of a bias that negatively impacts feature engineering.

## 2 RELATED WORK

**Feature Engineering Without Large Language Models.** Previous work dedicated considerable effort toward automating the process of feature engineering (Kanter & Veeramachaneni, 2015; Prado & Digiampietri, 2020; Mumuni & Mumuni, 2024). Various black-box methods have been proposed, such as ExploreKit (Katz et al., 2016), AutoFeat (Horn et al., 2020), BioAutoML (Bonidia et al., 2022), FETCH (Li et al., 2023), and OpenFE (Zhang et al., 2023). These methods typically generate new features in two steps: 1) create a large set of candidate features by applying mathematical (e.g., `Add`) or functional (e.g., `GroupByThenMean`) operators to features, and 2) return a small set of promising features selected from all candidate features.

**Feature Engineering with Large Language Models.** LLMs allow us to exploit their (potential) domain knowledge for feature engineering. LLMs can act as a proxy to a domain expert or data scientist during the feature engineering process. To illustrate, LLMs can be prompted to suggest code for generating new features (Hollmann et al., 2024; Hirose et al., 2024), to select predictive features (Jeong et al., 2024), or to use rule-based reasoning for generation (Nam et al., 2024). Most notably, CAAFE (Hollmann et al., 2024) presents a simple yet effective method to generate new features by proposing Python code to transform existing features in the dataset into new valuable features. This method lays the foundation to our proposed feature generation method, due to its wide application in practice[2] and also in research (Malberg et al., 2024; Guo et al., 2024; Zhang et al., 2024b).

**Bias in Large Language Models.** LLMs exhibit explicit and implicit biases. Explicit biases can be, among others, gender or racial discrimination in generated text (Kotek et al., 2023; Navigli et al., 2023; Gallegos et al., 2024; Bang et al., 2024). Moreover, LLMs can have implicit biases, such as specific words and phrases frequently re-used in generated text. As a result, Liang et al. (2024) were able to use recent trends in word frequency to detect and analyze LLM-generated text. Our method is similar to the investigation by Liang et al. (2024) but focuses on *operator* instead of word frequency.

**Other Applications of Large Language Models for Tabular Data.** Many researchers recently started using LLMs for applications related to tabular data. To avoid confusion in this plethora of recent work, we highlight similar but not directly related work to our contribution. Our work is not directly related to LLMs to tabular question answering (Ghosh et al., 2024; Grijalba et al., 2024; Wu et al., 2024), tabular dataset generation (Borisov et al., 2022b; van Breugel et al., 2024; Panagiotou

---

[2]For example, see these recent Kaggle competition write-ups (Hatch, 2024; Türkmen, 2024).

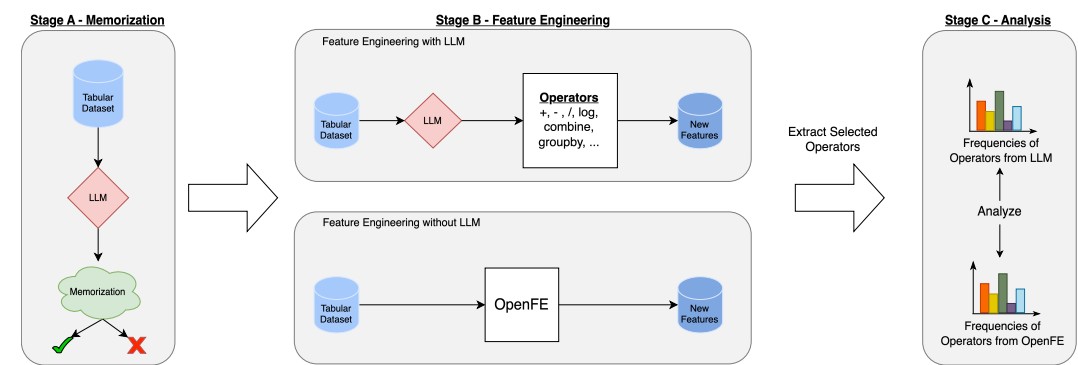

Figure 1: **Our Method to Analyze Feature Engineering Bias of an LLM.** Our method is split into three stages. **In the first stage A)**, we discard tabular datasets, which the LLM might have memorized. **In the second stage B)**, we instruct the LLM to select the best operators to engineer new features. At the same stage, we use OpenFE (Zhang et al., 2023), black-box automated feature engineering, to determine a proxy for the optimal operators. **In the third stage C)**, we compare the frequencies of operators for the LLM and OpenFE. That is, we look for anomalies when contrasting the distributions, such as using simple operators much more than complex ones.

et al., 2024), tabular data manipulation (Zhang et al., 2024a; Qian et al., 2024; Lu et al., 2024), or tabular few-shot predictions (Hegselmann et al., 2023; Han et al., 2024; Gardner et al., 2024)

## 3 METHOD: ANALYZING FEATURE ENGINEERING BIAS

We propose a three-stage method to assess the bias of an LLM when used to engineer new features for tabular data problems. Our three-stage method **A)** tests the LLM for memorization of benchmark datasets; **B)** engineers new features with an LLM as well as black-box automated feature engineering; and **C)** analyzes the bias of the LLM. We visualize our method in Figure 1.

**A) Memorization Test.** Given that language models are trained on vast amounts of publicly available data, we must account for the possibility that the LLM memorizes a dataset and optimal new features prior to our evaluation. To mitigate the risk of dataset-specific bias influencing the LLM during feature engineering, we test the LLM for memorization of datasets using the methods proposed by Bordt et al. (2024). Specifically, we conducted the row completion test, feature completion test, and the first token test. We consider a success rate of $50\%$ or higher in any test an indicator that the evaluation of the dataset is biased. In such cases, the dataset is excluded from further evaluation.

**B) Feature Engineering.** We propose a straightforward and interpretable feature engineering method for LLMs. Given a dataset, the LLM is supplied with context information, including the name and description. In addition, a comprehensive list of all features and critical statistical information for each feature (e.g., datatype, number of values, minimum, maximum, etc.) are provided. The instructions prompt also contains a pre-defined set of operators for engineering new features, each with a description indicating whether an operator is unary (applicable to one feature) or binary (requiring two features). We detail our full prompting specifications with examples in Appendix A. The LLM is then instructed to generate precisely one new feature by selecting one or two existing features and applying one of the available operators. We employ chain-of-thought (CoT) prompting (Wei et al., 2023) to boost the expressive power of the LLM (Li et al., 2024). In addition, the LLM explains why a feature was generated with CoT. We also employ a feedback loop, similar to CAAFE (Hollmann et al., 2024), which we detail in Appendix A.

Our approach to feature engineering with LLMs deviates from prior work (Hollmann et al., 2024; Hirose et al., 2024; Jeong et al., 2024) because we do not rely on code generation. This might put LLMs at a disadvantage because we reduce their potential expressiveness. However, we see this disadvantage outweighed by three significant advantages: first, we (almost) nullify the failure rate of generated code, which can be as much as $95.3\%$ for small models (Hirose et al., 2024) ; second,

we can control which operators the LLM uses; and third, we can extract applied operators from structured output without a (failure prone) code parser – enabling our study.

Fundamentally, we aim to compare the distribution over operators suggested by the LLM to the distribution over the optimal operators. That said, we do not know the optimal operators for a dataset. Thus, as a proxy, we use the operators suggested by OpenFE (Zhang et al., 2023).
To the best of our knowledge, OpenFE is the most recent, well-performing, and highly adopted[3] automated feature engineering tool. OpenFE suggests a set of new features after successively pruning *all possible new features* generated by a set of operators. To do so, OpenFE uses multi-fidelity feature boosting and computes feature importance.

**C) Analysis.** Finally, we analyze bias in feature engineering with LLMs using trends in the frequencies of operators. Therefore, we save the operators used by LLMs and black-box automated feature engineering from the previous stage. Subsequently, we compute the distribution over the frequencies of operators. This database allows us to visualize, inspect, and contrast the functional behavior of feature engineering with LLMs.

## 4 EXPERIMENTS

We extensively evaluate the bias of 4 LLMs for 21 operators across 27 classification datasets.

**Large Language Models.** We use four LLMs hosted by external providers via APIs. In detail, we used *GPT-4o-mini* (OpenAI, 2024) and *Gemini-1.5-flash* (Gemini-Team, 2024) to represent big frontier LLMs and *Llama3.1 8B* (Touvron et al., 2023) and *Mistral 7B Instruct v0.3* (Jiang et al., 2023), hosted by Together AI[4], to represent small open-source models. The API usage cost $\sim$200\$.

**Operators.** In this study, we use a fixed set of applicable operators. These operators represent a subset of the operators provided by OpenFE. We categorize the available operators into *simple* and *complex* operators. *Simple operators* apply straightforward arithmetic operations, such as adding two features. Furthermore, these operators are characterized by their relatively low computational complexity, typically $O(n)$. In contrast, *complex operators* perform more advanced transformations, such as grouping or combining the existing data into distinct subsets, followed by various aggregation functions. Compared to *simple operators*, *complex operators* generally exhibit a higher computational complexity of $O(n \log n)$ or greater. We present all operators and their categories in Appendix B.

**Datasets.** We used 27 out of 71 classification datasets from the standard AutoML benchmark (Gijsbers et al., 2024), which consists of curated tabular datasets from OpenML (Vanschoren et al., 2014). First, to avoid too large input prompts as well as extensive compute requirements, we selected all datasets with up to 100 features, 100 000 samples, and 10 classes – resulting in 36 datasets. We had to remove the `yeast` dataset due to insufficient samples per class for 10-fold cross-validation. Of the remaining 35 available datasets, 8 ($\sim$23%) failed our memorization tests (see Appendix E) with at least one LLM, making them unsuitable for further evaluation – resulting in 27 datasets.

**Evaluation Setup.** For each dataset, we perform 10-fold cross-validation. For each fold, we run OpenFE and prompt each LLM to generate 20 features. We assess the predictive performance of feature engineering following Zhang et al. (2023) by evaluating LightGBM (Ke et al., 2017) on the original features and the original features plus the newly generated features. Note, due to using a feedback loop, we only add features to the dataset when they improve the predictive performance on validation data (see Appendix A) We measured predictive performance using ROC AUC. Moreover, we mitigate a *positional bias* of our prompt template by repeating feature generation five times with an arbitrary order of operators. Finally, we compute the frequencies of operators across all new features, in total 27 000.

## 5 RESULTS

We order our results as follows: first, we demonstrate that a bias exists; then, we show that the bias negatively impacts the performance of feature engineering; and finally, we rule out confounding factors of the prompt template with an additional experiment.

---

[3]At the time of writing, OpenFE's GitHub repository has $\sim$760 stars and $\sim$100 forks.
[4]https://www.together.ai/

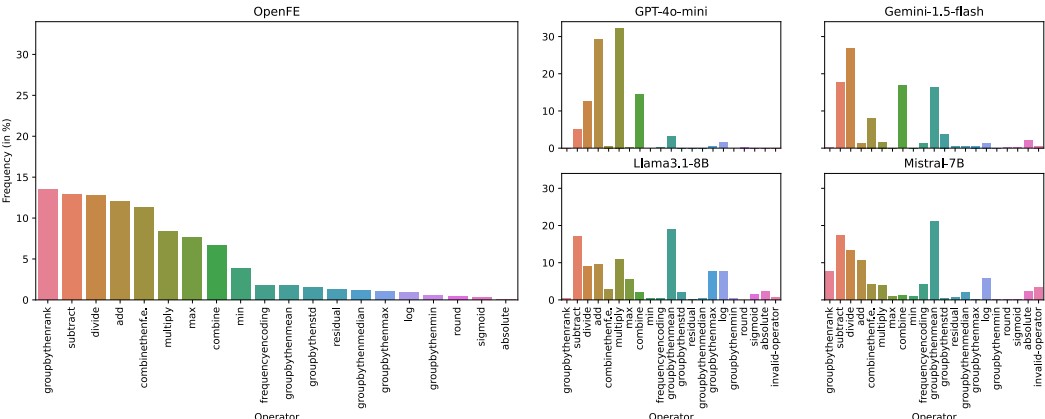

Figure 2: **Frequency of Feature Engineering Operators.** We present the frequency of how often an operator was used to create a new feature across all datasets, folds, and repetitions for OpenFE, *GPT-4o-mini*, *Gemini-1.5-flash*, *Llama3.1 8B*, and *Mistral 7B*. The highest frequency observed for OpenFE is ∼13.42% with the complex operator GroupByThenRank. In contrast, for *GPT-4o-mini*, it is ∼32.27% with the simple operator Multiply.

Table 1: **Names and Frequencies of the Most and Least Frequent Operator.** We present the most frequent (Max Freq.) and least frequent (Min Freq.) operator and their frequency per method/model. Each LLM model has a higher maximal and lower minimal frequency than OpenFE.

| Method/Model | Operator (Max Freq.) | Frequency (in %) | Operator (Min Freq.) | Frequency (in %) |
|---|---|---|---|---|
| OpenFE | groupbythenrank | **13.42** | absolute | **0.11** |
| GPT-4o-mini | multiply | 32.27 | min/groupbythenmean | **0.00** |
| Gemini-1.5-flash | divide | 26.87 | min | 0.02 |
| Llama3.1-8B | groupbythenmean | 18.96 | round | **0.00** |
| Mistral-7B-v0.3 | groupbythenmean | 21.13 | round | 0.09 |

**HYPOTHESIS 1:** FEATURE ENGINEERING WITH LARGE LANGUAGE MODELS IS BIASED TOWARD SIMPLE OPERATORS.

Figure 2 illustrates the operators' frequencies for OpenFE and the four LLMs. None of the language models replicate the distribution found by OpenFE. Although, notably, the distribution of *Llama3.1 8B* and *Mistral 7B* appear most similar. This discrepancy is particularly noticeable for the most frequently used *complex operators* by OpenFE, GroupByThenRank and CombineThenFrequencyEnconding. Neither are among the 3 most frequent operators for any LLM.

Table 1 presents names and frequencies of the most frequently generated operators by OpenFE and each LLM. Surprisingly, the small open-source LLMs most often select a complex operator, while both big LLMs favor simple operators. Nevertheless, the frequencies for the most used operators are significantly higher for LLMs than those observed for OpenFE. We further this analysis with Table 2, which shows all operators required to accumulate 90% of the total distribution. Notable, *GPT-4o-mini* features only five operators, with four of them - add, subtract, multiply, divide - representing basic arithmetic operators. This highlights a lack of complexity in the applied operators of one of the most complex LLMs.

In some cases, LLMs could not follow the instructions of our prompt template for generating a new feature. In these cases, the LLM usually proposed an operator not on the list of allowed operators. We show the frequency of occurrences for such invalid-operators in Figure 2, represented by the right-most operator. *Mistral-7B*, exhibits the highest frequency of invalid-operator across all LLMs, with the frequency even exceeding other allowed operators and being part of the ten most used operators for this LLM, as shown in Table 2. While concerning, failures for code-generation-based feature engineering methods of a similar model size are still much higher; see (Hirose et al., 2024).

Table 2: **Operators Making Up** 90% **of the Total Distribution.** We show the set of operators that make up 90% of the frequency distribution. OpenFE and both small open-source models require 10 operators to obtain 90% while *GPT-4o-mini* takes only 5.

| Model | Operators | Count | Cumulative Frequency (in %) |
|---|---|---|---|
| OpenFE | groupbythenrank, subtract, divide, add combinethenf.e., multiply, max, combine, min, frequencyencoding | **10** | 90.40 |
| GPT-4o-mini | multiply, add, combine, divide, subtract | 5 | **93.63** |
| Gemini-1.5-flash | divide, subtract, combine, groupbythenmean, combinethenf.e., groupbythenstd, absolute | 7 | 91.68 |
| Llama3.1-8B | groupbythenmean, subtract, multiply, add, divide, log, groupbythenmax, max, combinethenf.e., absolute | **10** | 91.62 |
| Mistral-7B-v0.3 | groupbythenmean, subtract, divide, add, groupbythenrank, log, frequencyencoding, combinethenf.e., multiply, invalid-operator | **10** | 91.23 |

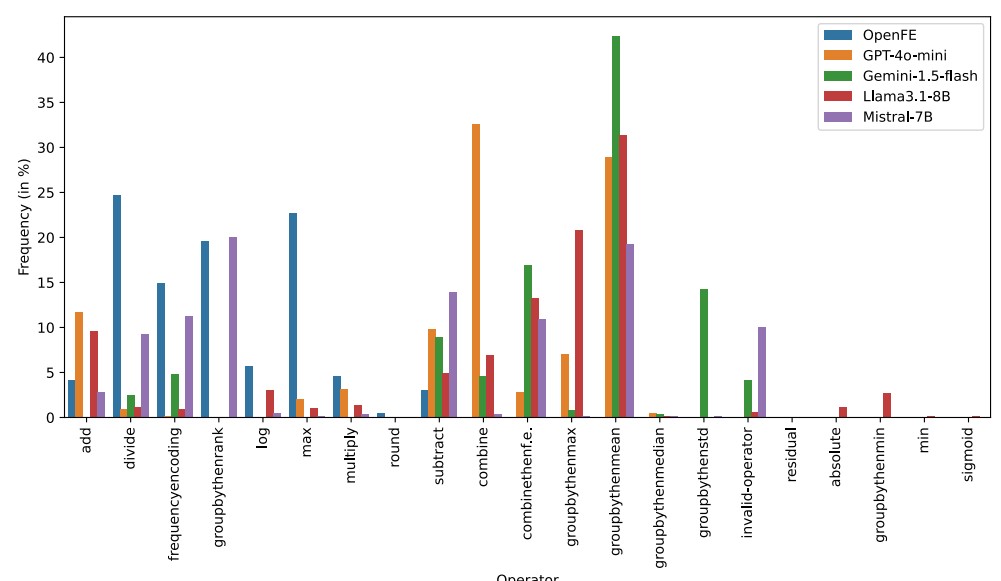

Figure 3: **Frequency of Feature Engineering Operators for a Dataset With Strong Bias.** We visualize the frequencies of operators for the "Amazon_employee_access" dataset. OpenFE favors operators to the left, while the LLMs focus on the operators to the right. For this dataset, LLMs favor complex operators (e.g., GroupByThenMean) but also frequently suggest Add and Subtract. OpenFE suggest simple operators most often but also GroupByThenRank. *Mistral-7B* can match the suggestions of OpenFE better than other LLMs.

We analyzed the bias of LLMs across a collection of datasets to obtain a meaningful conclusion. However, practitioners likely want to know if an LLM exhibits a bias for their data. Therefore, we highlight two of the 27 datasets as an example for practitioners. Figure 3 shows the operator frequencies for the "Amazon_employee_access" dataset. We observe that the LLMs exhibit a very different distribution from OpenFE, indicating a strong bias of LLMs. While the features engineered by the LLMs match each other, only *Mistral-7B* engineers features similar to OpenFE. We highlighted this dataset because it was the only one where LLMs seem to favor complex features more than simple ones. This indicates that the dataset, and its information presented in the prompt, contribute to the bias exhibited by LLMs. For the second example, we show the operator frequencies for the "phoneme" dataset in Figure 4. The features engineered by the LLMs were also found to be optimal operators by OpenFE. However, *GPT-4o-mini* strongly prefers Add, similar to the bias observed across all datasets. Yet, the LLMs rarely use the one complex operator frequently found by OpenFE (Resiudal). This shows that an analysis across datasets is required to avoid dataset-specific noise.

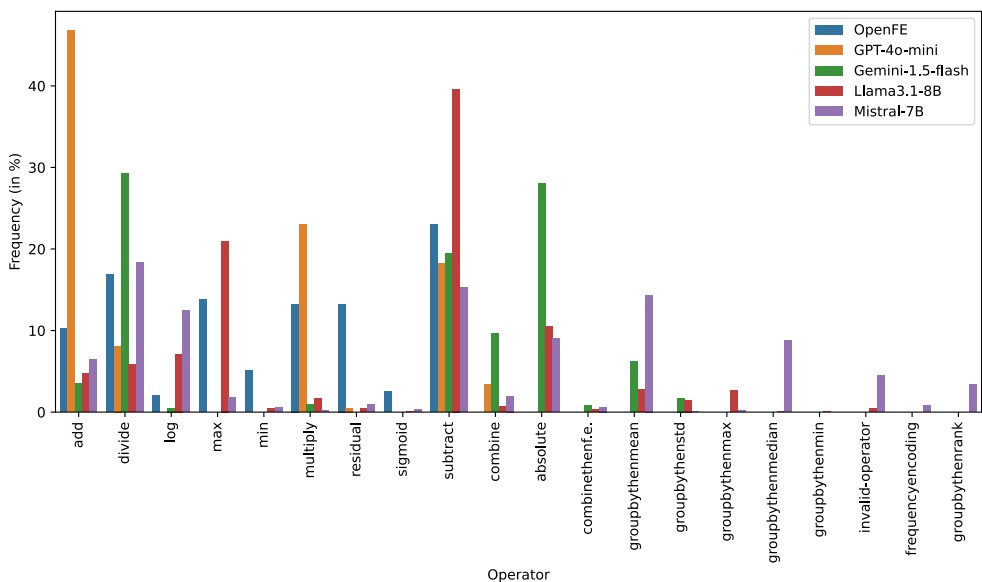

Figure 4: **Frequency of Feature Engineering Operators for a Dataset Without Strong Bias.** We visualize the frequencies of operators for the "phoneme" dataset. The distribution of the LLMs and OpenFE is reasonably well aligned, except for *GPT-4o-mini* for `Add`. OpenFE and all LLMs frequently create new features with `Divide` and `Subtract`. Nevertheless, the LLMs still fail to use the complex operator `Residual`, as done by OpenFE.

**HYPOTHESIS 2:** THE BIAS OF LARGE LANGUAGE MODELS NEGATIVELY IMPACTS FEATURE ENGINEERING.

Figure 5 and Table 3 present the relative improvements in predictive accuracy of the features engineered with each LLM and OpenFE in comparison to a system without feature engineering. We present raw results per dataset in Appendix F. The results demonstrate the negative impact of the bias on the quality of the generated features. The two big frontier models, which are more biased toward simple operators, perform worse on average. While the two small open-source models improve performance, but still perform worse than OpenFE. Additionally, the effectiveness of OpenFE is highlighted, improving predictive accuracy on 21 of 27 benchmark datasets.

Table 3: **Predictive Performance Improvement With Feature Engineering.** We show the number of datasets with improvements and the average relative improvement for OpenFE and each LLM.

| Method/Model | Improvements | Average Relative Improvement (in %) |
|---|---|---|
| OpenFE | **21/27** | **+0.638** |
| GPT-4o-mini | 10/27 | −0.507 |
| Gemini-1.5-flash | 6/27 | −1.161 |
| Llama3.1-8b | 16/27 | +0.165 |
| Mistral-7b-v0.3 | 14/27 | +0.164 |

As observed in Figure 5, adding the features engineered by *Gemini-1.5-flash* and *GPT-4o-mini* to the data performs much worse than no feature engineering for several outliers. This is particularly interesting because of the feedback loop we implemented. Our feedback loop only adds features to the data when it improves predictive performance on validation data. Therefore, these results suggest that both big frontier models sometimes engineer features that do not generalize to test data, even when their expressiveness is limited to a set of pre-defined operators. We manually investigated all datasets with a relative improvement above or below 2% for a pattern in the generated features. Compared to the overall distribution, these cases were not dominated by individual operators or

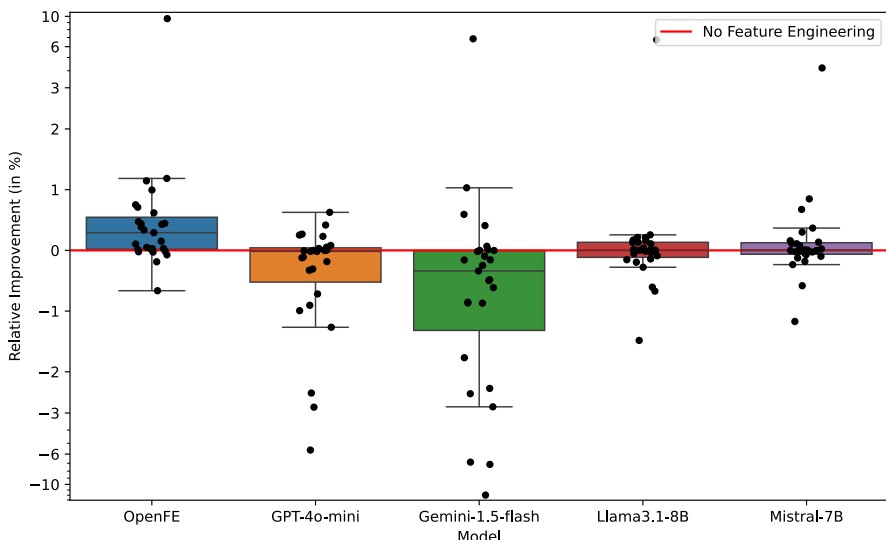

Figure 5: **Relative Improvements of Feature Engineering.** We visualize the distribution of relative improvements using boxplots for OpenFE and each LLM. We compute improvement relative to a LightGBM classifier trained only on the original dataset (red horizontal line). A higher relative improvement indicates that the performance of LightGBM improved when training on the original data plus the new features generated by OpenFE or an LLM. OpenFE improves the performance on most datasets and has the highest median relative improvement, as shown by the black horizontal line in the box. In contrast, *Gemini-1.5-flash* rarely improves the performance.

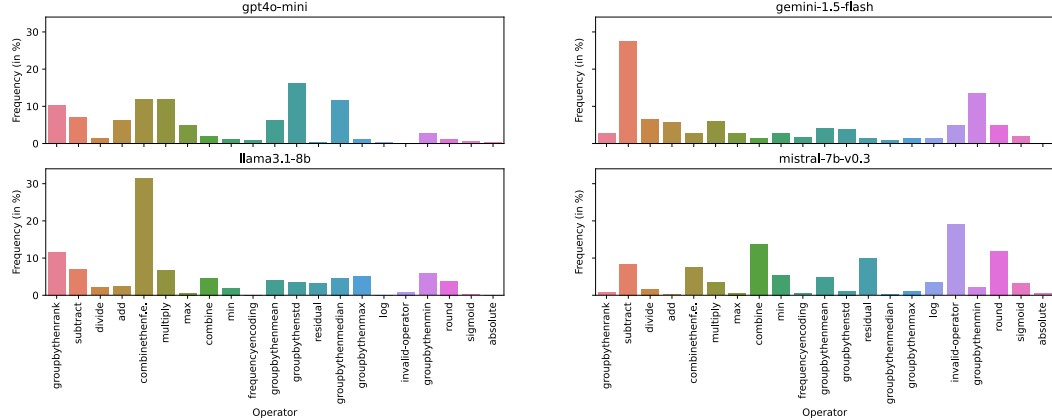

Figure 6: **Frequency of Feature Engineering Operators for Random Search with LLMs.** Distribution over the frequency of feature engineering operators when simulating random search by masking the operators' names in the prompt. The frequency denotes how often an operator was used to create a new feature across all datasets, folds, and repetitions for *GPT-4o-mini*, *Gemini-1.5-flash*, *Llama3.1 8B*, and *Mistral 7B*. Compared to Figure 2, the distribution for all models exhibits a significantly higher degree of uniformity, and the previously observed bias toward simple operators does not manifest. Notably, *Mistral 7B* has again the highest frequency of generating invalid operators.

specific operator types. We leave it to future work to investigate how well the feature engineering of LLMs generalizes to unseen data.

**ADDITIONAL EXPERIMENT.** ENGINEERING RANDOM FEATURES WITH LLMS.

We additionally investigate whether the observed bias primarily arises from positional preferences related to the positioning of operator names in our prompt. We additionally investigate whether our prompt template influenced our results. Therefore, we adapt our prompt template to mirror a random search. That is, we force an LLM to generate new features by randomly selecting the most appropriate operator. To this end, we employed the same experimental setup as our primary experiments. However, we masked the actual names of the operators in the instructions prompt. Each operator is assigned a numeric label, which the LLM selects. Subsequently, these numeric labels are mapped back to the names of the corresponding operators. Notably, we again shuffle the order of operators five times.

Figure 6 shows the distribution over the frequency of operators with LLM-based random search for feature engineering. We observe that all models exhibit a significantly higher degree of uniformity compared to Figure 2. Yet, we do not observe total uniformity as expected for a random search, which aligns with observations for LLMs by Hopkins et al. (2023). Moreover, the previously observed bias toward simple operators does not manifest anymore. This is particularly visible for *GPT-4o-mini*. *Mistral 7B* has again the highest selection frequency of invalid operator labels, i.e., fails to follow the prompt's instructions. We conclude that the positioning the operator names in our prompt template did not cause the bias toward simple operators. Instead, the content of the prompt, in combination with the LLM, causes the bias.

## 6 CONCLUSION

In this work, we propose a method to evaluate whether large language models (LLMs) are biased when used for feature engineering for tabular data. Our method detects a bias based on anomalies in the frequency of operators used to engineer new features (e.g., `Add`). In our experiments, we evaluated the bias of four LLMs. Our results reveal a bias towards simpler operators when engineering new features with LLMs. Moreover, this bias seems to negatively impact the predictive performance when using features generated by an LLM.

In conclusion, the contributions of our work are a method to detect bias in LLMs and evidence that a bias toward simple operators exists. The findings of this method underscore the necessity to further strengthen LLMs to truly unlock their potential for tabular data problems. Our work underscores the importance of developing methods to mitigate bias in downstream applications. Promising methods for future work to explore are in-context learning (e.g., prompt tuning) or fine-tuning the LLM to favor optimal operators. In the long term, after identifying and addressing biases in LLMs, we can fully liberate ourselves from manual feature engineering. This will allow us to leverage LLMs as reliable and efficient automated data science agents for tabular data.

**Limitations and Broader Impact.** Our study on the bias of LLMs still has limitations because it is the first of its kind for automated data science. We detect a bias but do not support practitioners to determine why an LLM might be biased. Similarly, given how frontier LLMs are trained and deployed, we focused on assessing bias in the models' outputs rather than internal mechanisms. Lastly, our study is limited to four LLMs, while practitioners can also choose from many other potentially biased LLMs. Our findings and proposed method do not create any negative societal impacts. Instead, both can have positive societal impacts because they increase our understanding and (mis)trust when using large language models for feature engineering.

**Reproducibility Statement** We made the code used in our experiments publicly available at [LINK REDACTED DUE TO ANONYMITY] to ensure our work's reproducibility and enable others to analyze their LLM for bias. Furthermore, we used public datasets from OpenML (Vanschoren et al., 2014). The appendix details how we interact with the LLMs, which model versions we use, and the results of our memorization tests. Furthermore, we present non-aggregated results in the appendix.

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

## A    FULL PROMPTING SPECIFICATIONS

### A.1    PROMPT TEMPLATES

Figure 7 contains the system prompt we used to interact with the LLM during feature engineering. Figure 8 contains an example instruction prompt for the *blood-transfusion-service-center* dataset. First, we include a list of statistical values that describe each feature. Secondly, a list of all allowed operators as defined in B is passed with a description for whether the operator is a binary operator and thus applicable to two features at a time or unary, only applicable to one feature. Finally, we will give strict formatting instructions for the expected output of the LLM. We demand that each newly proposed feature contain four elements. First, we require reasoning as to why the given feature was selected. Second, we require a combination of exactly one operator and one or two existing features (depending on whether the chosen operator is a unary or a binary operator). Finally, we request a name for the new feature and a short description of its contents in the context of the dataset.

---

You are an expert data scientist performing effective feature engineering on a dataset. You will get a short description of every feature in the dataset. This description will contain some statistical information about each feature.

Example of the information you will get about a feature: Feature 1: Type: int64, Feature size: 100, Number of values: 100, Number of distinct values: 100, Number of missing values: 0, Max: 100, Min: 0, Mean: 50, Variance: 100, Name: name. Sample: First couple of rows of the dataset.

If some value carries the value EMPTY, it means that this value is not applicable for this feature. You are also provided a list of operators. There are unary and binary operators. Unary operators take one feature as input and binary operators take two features as input.

Example of the information you will get about an operator: Operator: SomeOperator, Type: Binary You are now asked to generate a new feature using the information from the features and the information from the operators as well as your own understanding of the dataset and the given domain. There will be an example of how your response should look like. You will only answer following this example. Your response will containing nothing else. You are only allowed to select operators from the list of operators and features from the list of features. Your are only allowed to generate one new feature. Your newly generated feature will then be added to the dataset.

---

Figure 7: **Our System Prompt.** The contents of the full system prompt, which is sent to the LLM before the instructions to generate new features for a given dataset.

### A.2    FEEDBACK LOOP

We additionally employ a feedback loop to supply the LLM with additional knowledge about its generated features, similar to CAAFE (Hollmann et al., 2023). After the LLM proposes a feature, the new feature is manually computed by applying the requested operator to the respective features. Before a feature is added to the dataset, we test whether it yields improvements in ROC AUC on the given dataset to prevent the addition of noisy features. Each proposed feature is added to the dataset, and 10-fold cross-validation is conducted using LightGBM (Ke et al., 2017). When compare the average ROC AUC score to the average ROC AUC scores over the dataset without the new feature. The new feature is subsequently only added to the dataset if it improves the average ROC AUC score. In the next feature generation request, the user prompt additionally contains information about the previous feature, including its name, description, reasoning, and the actual transformation request from the prior round. We also pass the change in ROC AUC score that the last feature yielded in comparison to the dataset without the new feature.

## B    LIST OF OPERATOR AND THEIR CATEGORIES

See Tables 4 and 5 for an overview of operators used in our study.

## C    DATASETS

See Table 6 for an overview of all dataset used in our study.

Feature 1: Type: int64, Feature size: 673, Number of values: 673, Number of distinct values: 30, Number of missing values: 0, Max: 74, Min: 0, Mean: 9.543, Variance: 67.016, Name: V1
Feature 2: Type: int64, Feature size: 673, Number of values: 673, Number of distinct values: 33, Number of missing values: 0, Max: 50, Min: 1, Mean: 5.558, Variance: 35.916, Name: V2
Feature 3: Type: int64, Feature size: 673, Number of values: 673, Number of distinct values: 33, Number of missing values: 0, Max: 12500, Min: 250, Mean: 1389.673, Variance: 2244785.309, Name: V3
Feature 4: Type: int64, Feature size: 673, Number of values: 673, Number of distinct values: 77, Number of missing values: 0, Max: 98, Min: 2, Mean: 34.358, Variance: 599.813, Name: V4
Operator: FrequencyEncoding, Type: Unary
Operator: Absolute, Type: Unary
Operator: Log, Type: Unary
Operator: SquareRoot, Type: Unary
Operator: Sigmoid, Type: Unary
Operator: Round, Type: Unary
Operator: Residual, Type: Unary
Operator: Min, Type: Binary
Operator: Max, Type: Binary
Operator: Add, Type: Binary
Operator: Subtract, Type: Binary
Operator: Multiply, Type: Binary
Operator: Divide, Type: Binary
Operator: Combine, Type: Binary
Operator: CombineThenFrequencyEncoding, Type: Binary
Operator: GroupByThenMin, Type: Binary
Operator: GroupByThenMax, Type: Binary
Operator: GroupByThenMean, Type: Binary
Operator: GroupByThenMedian, Type: Binary
Operator: GroupByThenStd, Type: Binary
Operator: GroupByThenRank, Type: Binary
Here is an example of how your return will look like. Suppose you want to apply operator A to Feature X and Feature Y. Even if you know the names of features X and Y you will only call them by their indices provided to you. You will not call them by their actual names. You will return the following and nothing else: REASONING: Your reasoning why you generated that feature.; FEATURE: A(X, Y); NAME: name; DESCRIPTION: This is the feature called name. This feature represents ... information.

Figure 8: **Our Instruction Prompt.** The contents of the instruction prompt on the example of the *blood-transfusion-service-center* dataset. This prompt is send every time the LLM is instructed to generate a feature for a given dataset. The order of operators is shuffled according to the explanations in Section 4.

## D  LARGE LANGUAGE MODELS

See Table 7 for an overview of the specific model versions used in this study.

## E  MEMORIZATION TEST RESULTS

To mitigate the risk of dataset-specific bias, we conduct memorization tests (Bordt et al., 2024) before our experiment. From the forms proposed Bordt et al. (2024) for dataset understanding by a large language model (LLM), we consider actual memorization of the dataset to be most influential to our evaluation. Therefore, we employ the tests that evaluate the level to which extend a LLM memorizes a given datasets. These tests include a (1) row completion test, (2) feature completion test, and (3) first token test. Each test prompts a given LLM with 25 different samples from the dataset. We consider a success rate of $50\%$ on at least one test as an indicator of memorization. If one of the four different used language models implied signs of memorization, the tests where not further conducted

Table 4: **Simple Operators.**

| Operators |
| --- |
| abs |
| log |
| sqrt |
| round |
| min |
| max |
| add |
| subtract |
| multiply |
| divide |

Table 5: **Complex Operators.**

| Operators |
| --- |
| residual |
| sigmoid |
| frequencyencoding |
| groupbythenmin |
| groupbythenmax |
| groupbythenmean |
| groupbythenmedian |
| groupbythenstd |
| groupbythenrank |
| combine |
| combinethenfrequencyencoding |

for the other remaining models. Table 8 and Table 9 present the results for the memorization tests for all initial datasets on all four models.

## F    PREDICTIVE ACCURACY RESULTS

See Table 10 to see the average ROC AUC scores for all folds for all datasets for each method.

## G    ADDITIONAL EXPERIMENTS

To further solidify the results of our study we conducted some additional experiments.

### G.1    STATISTICAL SIGNIFICANCE

We test the statistical significance of our results on predictive accuracy across all benchmark datasets and methods. The results of these tests are presented in the critical difference diagram in Figure 9. As visible in this diagram, OpenFE has the highest rank across all methods, outperforming all LLM-based methods. GPT-4o-mini and Gemini-1.5-flash exhibit no statistical difference and are last in ranks, matching our findings from Table 3 and Figure 5. Further, the similarity in performance between Llama3.1-8b and Mistral-7b-v0.3 is further solidified.

### G.2    BIAS IN GPT-4O

To evaluate whether the apparent bias can be fixed by selecting more powerful models as foundation, we conducted the same experiments on a subset of the benchmark datasets in OpenAI's GPT-4o model. We compared the distribution of selected operators by GPT-4o to the distribution of GPT-4o-mini, presented in Figure 10. When considering the churn and phoneme dataset, a strong similarity between the distributions of GPT-4o and GPT-4o-mini is apparent. For the ada and shuttle dataset the distribution of GPT-4o is visibly smoother in comparison to the distribution of GPT-4o-mini. However when considering the types of selected features, the bias towards simple operators is still very strongly represented by GPT-4o, indicating that the usage more powerful models does not fix the bias towards simpler operators.

### G.3    FEATURE SELECTION FREQUENCIES

We evaluate the frequencies of selected features per dataset. For each dataset we calculate the frequencies with which each feature is selected over all feature generation steps. As presented in Figure 11 the LLM is relatively certain which features to select, indicated by high frequencies for few features on most datasets.

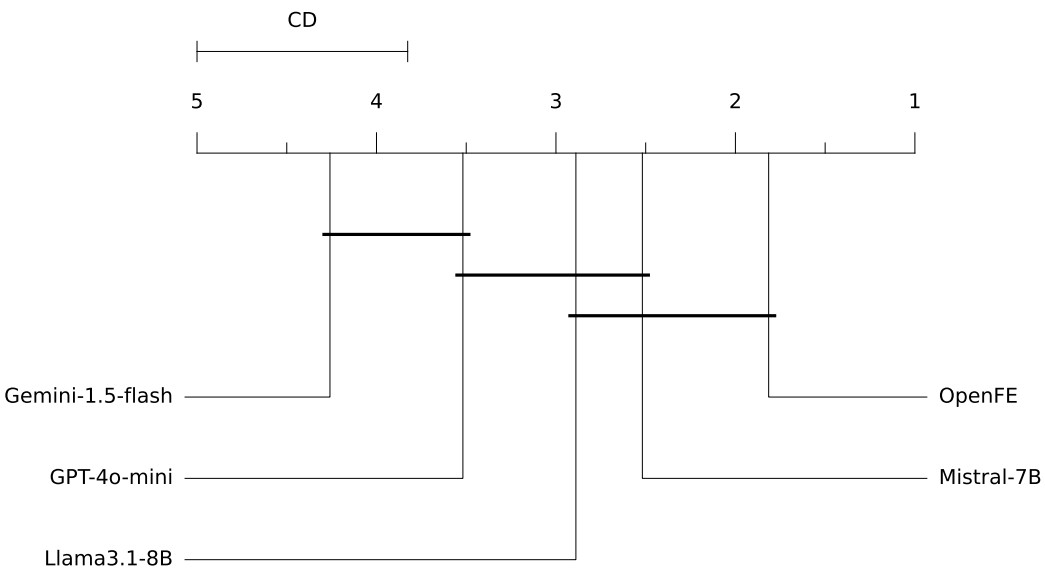

Figure 9: **Critical Difference Plots for Test Scores.** Mean rank of the methods (lower is better). Methods connected by a bar are not significantly different.

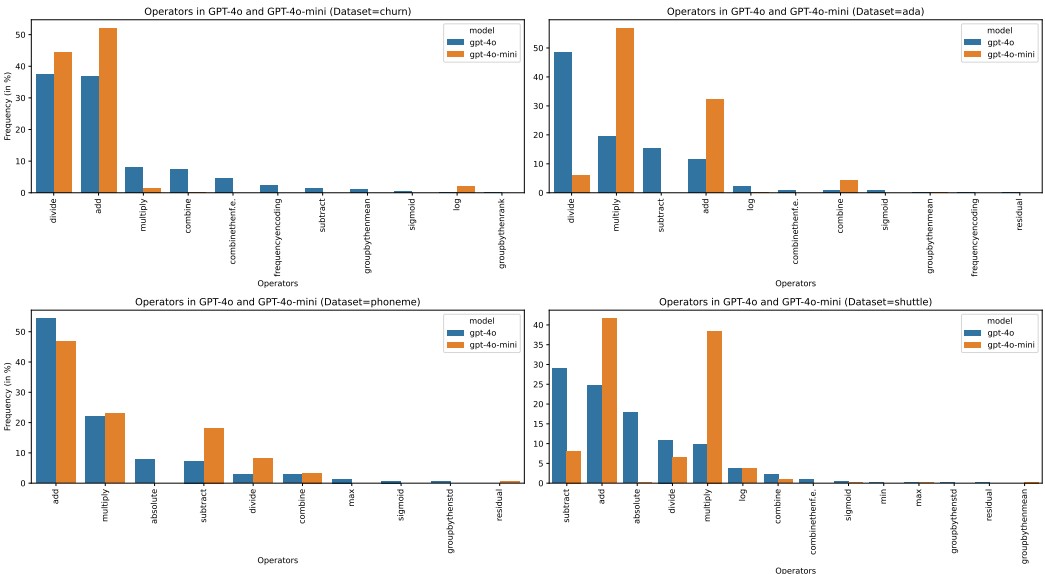

Figure 10: **Comparison of Operator Distributions for GPT-4o and GPT-4o-mini**. We present the distributions of selected operators on 4 different benchmark datasets. Generally the distribution for GPT-4o is smoother in comparison to GPT-4o-mini. However, the problem of relying heavily on only few (simple) operators, is similar to GPT-4o-mini, as visible from the set of selected operators as well as the frequencies with which the simple operators are selected accross all 4 datases.

Table 6: **Benchmark Datasets** The table contains all datasets from the AutoML benchmark (Gijsbers et al., 2024) that we used in our experiments. We selected dataset based on our constraints defined in Section 4. All datasets listed in this Table were tested for memorization of the LLMs.

| Datastet ID | Dataset | Features | Samples | Classes |
|---|---|---|---|---|
| 190411 | ada | 49 | 4147 | 2 |
| 359983 | adult | 15 | 48842 | 2 |
| 359979 | amazon_employee_access | 10 | 32769 | 2 |
| 146818 | australian | 15 | 690 | 2 |
| 359982 | bank-marketing | 17 | 45211 | 2 |
| 359955 | blood-transfusion-service-center | 5 | 748 | 2 |
| 359960 | car | 7 | 1728 | 4 |
| 359968 | churn | 21 | 5000 | 2 |
| 359992 | click_prediction_small | 12 | 39948 | 2 |
| 359959 | cmc | 10 | 1473 | 3 |
| 359977 | connect-4 | 43 | 67557 | 2 |
| 168757 | credit-g | 21 | 1000 | 2 |
| 359954 | eucalyptus | 20 | 736 | 5 |
| 359969 | first-order-theorem-proving | 52 | 6118 | 6 |
| 359970 | gesturephasesegmentationprocessed | 33 | 9873 | 5 |
| 211979 | jannis | 55 | 83733 | 4 |
| 359981 | jungle_chess_2pcs_raw_endgame_complete | 7 | 44819 | 3 |
| 359962 | kc1 | 22 | 2109 | 2 |
| 359991 | kick | 33 | 72983 | 2 |
| 359965 | kr-vs-kp | 37 | 3196 | 2 |
| 167120 | numerai28.6 | 22 | 96320 | 2 |
| 359993 | okcupid-stem | 20 | 50789 | 3 |
| 190137 | ozone-level.8hr | 73 | 2534 | 2 |
| 359958 | pc4 | 38 | 1458 | 2 |
| 359971 | phishingwebsites | 31 | 11055 | 2 |
| 168350 | phoneme | 6 | 5404 | 2 |
| 359956 | qsar-biodeg | 42 | 1055 | 2 |
| 359975 | satellite | 37 | 5100 | 2 |
| 359963 | segment | 20 | 2310 | 7 |
| 359987 | shuttle | 10 | 58000 | 7 |
| 168784 | steel-plates-fault | 28 | 1941 | 7 |
| 359972 | sylvine | 21 | 5124 | 2 |
| 190146 | vehicle | 19 | 846 | 4 |
| 146820 | wilt | 6 | 4839 | 2 |
| 359974 | wine-quality-white | 12 | 4898 | 7 |

Table 7: **Model API Versions.** The full model versions as specified by the respective API provider.

| Model | Model Version |
|---|---|
| gpt-4o-mini | gpt-4o-mini-2024-07-18 |
| gemini-1.5-flash | gemini-1.5-flash-001 |
| llama3.1-8b | Meta-Llama-3.1-8B-Instruct-Turbo (FP8 Quantization) |
| mistral7b-v0.3 | Mistral-7B-Instruct-v0.3 (FP16 Quantization) |

Table 8: **Memorization Tests Results GPT and Llama.** We present the results of all three memorization tests (Bordt et al., 2024), (1) row completion test (r.c.), (2) feature completion test (f.c.) and first token test (f.t.) for *gpt-4o-mini* and *llama3.1-8b*. For each dataset and each test, the number of runs which imply signs of memorization are listed. Each test ran 25 tries per dataset. Sometimes, the LLM failed to match the expected outcome sequences required by the tests (noted as -). If a prior language model exhibited signs of memorization for a dataset, the tests were not further conducted for subsequent models (noted as X)

| Dataset | gpt4-r.c. | gpt4-f.c. | gpt4-f.t. | llama3.1-r.c. | llama3.1-f.c. | llama3.1-f.t. |
|---|---|---|---|---|---|---|
| ada | 0/25 | 0/25 | - | 0/25 | 0/25 | - |
| adult | 0/25 | 0/25 | 8/25 | 0/25 | 0/25 | 3/25 |
| amazon_employee_access | 0/25 | 0/25 | 6/25 | 0/25 | 0/25 | 5/25 |
| australian | 0/25 | 0/25 | 4/25 | 0/25 | 0/25 | 4/25 |
| bank-marketing | 0/25 | 0/25 | 11/25 | 0/25 | 2/25 | 5/25 |
| blood-transfusion... | 2/25 | 1/25 | 15/25 | X | X | X |
| car | 23/25 | 6/25 | 25/25 | X | X | X |
| churn | 0/25 | 0/25 | 4/25 | 0/25 | 0/25 | 3/25 |
| click_prediction_small | 0/25 | 0/25 | - | 0/25 | 1/25 | - |
| cmc | 0/25 | 0/25 | 13/25 | X | X | X |
| connect-4 | 0/25 | 5/25 | - | 0/25 | 5/25 | - |
| credit-g | 0/25 | 0/25 | 8/25 | 0/25 | 0/25 | 5/25 |
| eucalyptus | 0/25 | 0/25 | - | 0/25 | 0/25 | - |
| first-order-theorem-proving | 0/25 | 0/25 | - | 0/25 | 0/25 | 6/25 |
| gesturephase... | 0/25 | 0/25 | - | 0/25 | 0/25 | - |
| jannis | 0/25 | 0/25 | 10/25 | 0/25 | 0/25 | 8/25 |
| jungle_chess... | 20/25 | 1/25 | - | X | X | X |
| kc1 | 7/25 | 1/25 | 7/25 | 1/25 | 3/25 | 10/25 |
| kick | 0/25 | 0/25 | - | 0/25 | 0/25 | - |
| kr-vs-kp | 0/25 | 0/25 | - | 0/25 | 0/25 | - |
| numerai28.6 | 0/25 | 0/25 | 1/25 | 0/25 | 0/25 | 1/25 |
| okcupid-stem | 0/25 | - | 10/25 | 0/25 | - | 12/25 |
| ozone-level.8hr | 0/25 | 0/25 | - | 0/25 | 0/25 | 12/25 |
| pc4 | 0/25 | 3/25 | 7/25 | 0/25 | 0/25 | 5/25 |
| phishingwebsites | 0/25 | 1/25 | - | 0/25 | 1/25 | - |
| phoneme | 0/25 | 0/25 | 5/25 | 0/25 | 0/25 | 5/25 |
| qsar-biodeg | 0/25 | 0/25 | 1/25 | 0/25 | 0/25 | 3/25 |
| satellite | 0/25 | 1/25 | - | 0/25 | 2/25 | - |
| segment | 0/25 | 0/25 | - | 0/25 | 1/25 | - |
| shuttle | 0/25 | 0/25 | 9/25 | 0/25 | 4/25 | 8/25 |
| steel-plates-fault | 0/25 | 2/25 | 14/25 | X | X | X |
| sylvine | 0/25 | 0/25 | 7/25 | 0/25 | 0/25 | - |
| vehicle | 0/25 | 0/25 | 8/25 | 0/25 | 0/25 | 7/25 |
| wilt | 0/25 | 0/25 | 9/25 | 0/25 | 0/25 | 8/25 |
| wine-quality-white | 0/25 | 0/25 | 14/25 | X | X | X |
| yeast | 0/25 | 1/25 | 5/25 | 0/25 | 2/25 | 4/25 |

Table 9: **Memorization Tests Results Mistrial and Gemini.** We present the results of all three memorization tests (Bordt et al., 2024), (1) row completion test (r.c.), (2) feature completion test (f.c.) and first token test (f.t.) for *mistral7b-v0.3* and *gemini-1.5-flash*. For each dataset and each test the number of runs which imply signs of memorization are listed. Each test ran 25 tries per dataset. Sometimes, the LLM failed to match the expected outcome sequences required by the tests (noted as -). If a prior language model exhibited signs of memorization for a dataset, the tests were not further conducted for subsequent models (noted as X)

| Dataset | mistral7b-r.c. | mistral7b-f.c. | mistral7b-f.t. | gemini1.5-r.c. | gemini1.5-f.c. | gemini1.5-f.t. |
|---|---|---|---|---|---|---|
| ada | 0/25 | 0/25 | - | 0/25 | 0/25 | - |
| adult | 0/25 | 0/25 | 0/25 | 0/25 | 0/25 | 4/25 |
| amazon_employee_access | 0/25 | 0/25 | 0/25 | 0/25 | 0/25 | 5/25 |
| australian | 0/25 | 0/25 | 0/25 | 0/25 | 0/25 | 8/25 |
| bank-marketing | 0/25 | 0/25 | 0/25 | 0/25 | 0/25 | 10/25 |
| blood-transfusion... | X | X | X | X | X | X |
| car | X | X | X | X | X | X |
| churn | 0/25 | 0/25 | 0/25 | 0/25 | - | 7/25 |
| click_prediction_small | 0/25 | 0/25 | - | 0/25 | 0/25 | - |
| cmc | X | X | X | X | X | X |
| connect-4 | 0/25 | 2/25 | - | 0/25 | 2/25 | - |
| credit-g | 0/25 | 0/25 | 0/25 | 0/25 | 0/25 | 8/25 |
| eucalyptus | 0/25 | 0/25 | - | - | 0/25 | - |
| first-order-theorem-proving | - | 0/25 | - | 0/25 | 0/25 | - |
| gesturephase... | 0/25 | - | - | 0/25 | 0/25 | - |
| jannis | - | - | - | 0/25 | 0/25 | 12/25 |
| jungle_chess... | X | X | X | X | X | X |
| kc1 | 0/25 | - | 0/25 | 1/25 | 9/25 | 11/25 |
| kick | 0/25 | - | - | 0/25 | 0/25 | - |
| kr-vs-kp | 0/25 | - | - | 1/25 | 0/25 | - |
| numerai28.6 | - | - | - | 0/25 | 0/25 | 0/25 |
| okcupid-stem | 0/25 | - | 0/25 | - | - | 7/25 |
| ozone-level.8hr | - | - | - | 0/25 | 0/25 | 13/25 |
| pc4 | 0/25 | - | 0/25 | 0/25 | 4/25 | 14/25 |
| phishingwebsites | 0/25 | - | - | 0/25 | 0/25 | - |
| phoneme | 0/25 | - | 0/25 | 0/25 | 0/25 | 2/25 |
| qsar-biodeg | 0/25 | - | 0/25 | 0/25 | 0/25 | 3/25 |
| satellite | 0/25 | - | - | 0/25 | 6/25 | - |
| segment | 0/25 | - | - | 0/25 | 1/25 | - |
| shuttle | 0/25 | - | 0/25 | 0/25 | 1/25 | 8/25 |
| steel-plates-fault | X | X | X | X | X | X |
| sylvine | 0/25 | - | - | 0/25 | 0/25 | 11/25 |
| vehicle | 0/25 | - | 0/25 | 0/25 | 0/25 | 10/25 |
| wilt | 0/25 | - | 0/25 | 0/25 | 0/25 | 6/25 |
| wine-quality-white | X | X | X | X | X | X |
| yeast | 0/25 | - | 0/25 | 0/25 | 0/25 | 6/25 |

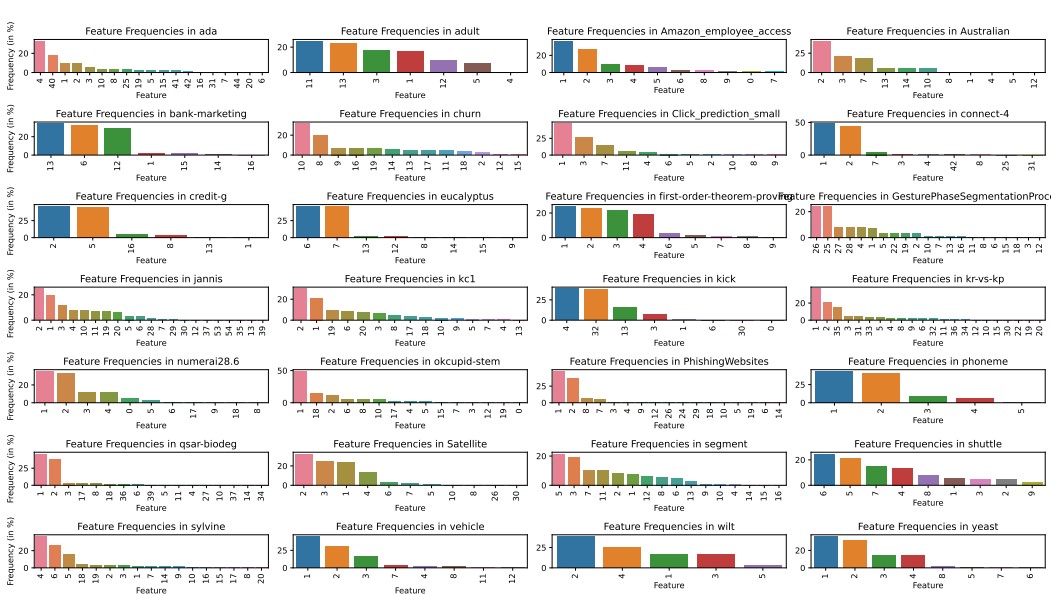

Figure 11: **Feature Selection Frequency per Dataset**. We present the frequencies with which each feature in a dataset is selected by the LLM over all feature generation steps. In most cases the LLM selects few features with a high frequency repeatedly.

Table 10: **Predictive Performance of Feature Engineering.** We show the average and standard deviation of the ROC AUC scores for all folds for all datasets. *Base* represents the baseline score without feature engineering. The scores for the four large language models additionally contain the average over all 5 shuffles of operator order in the instructions prompt. For each fold the respective method generated 20 new features. Features were only added to the method if it improved the feedback scores described in A.

| Dataset | Base | OpenFE | GPT-4o-mini | Gemini-1.5-flash | Llama3.1-8B | Mistral7B-v0.3 |
|---|---|---|---|---|---|---|
| ada | $0.912 \pm .016$ | $0.910 \pm .019$ | $0.910 \pm .019$ | $0.904 \pm .019$ | $0.911 \pm .017$ | $0.911 \pm .017$ |
| adult | $0.929 \pm .004$ | $\mathbf{0.931 \pm .004}$ | $0.929 \pm .004$ | $0.921 \pm .013$ | $\mathbf{0.929 \pm .004}$ | $\mathbf{0.929 \pm .004}$ |
| amazon_employee_access | $0.822 \pm .018$ | $\mathbf{0.825 \pm .017}$ | $0.776 \pm .038$ | $0.820 \pm .024$ | $\mathbf{0.823 \pm .019}$ | $\mathbf{0.825 \pm .017}$ |
| australian | $0.933 \pm .025$ | $0.932 \pm .021$ | $0.929 \pm .027$ | $0.927 \pm .019$ | $0.930 \pm .023$ | $0.931 \pm .028$ |
| bank_marketing | $0.935 \pm .007$ | $\mathbf{0.939 \pm .006}$ | $0.935 \pm .007$ | $0.934 \pm .007$ | $0.935 \pm .007$ | $0.935 \pm .007$ |
| churn | $0.923 \pm .028$ | $\mathbf{0.924 \pm .018}$ | $\mathbf{0.926 \pm .023}$ | $0.920 \pm .026$ | $\mathbf{0.924 \pm .025}$ | $\mathbf{0.924 \pm .026}$ |
| click_prediction_small | $0.607 \pm .014$ | $0.607 \pm .022$ | $0.602 \pm .019$ | $0.594 \pm .016$ | $0.603 \pm .017$ | $0.600 \pm .018$ |
| connect-4 | $0.876 \pm .004$ | $\mathbf{0.886 \pm .004}$ | $\mathbf{0.876 \pm .004}$ | $0.876 \pm .004$ | $\mathbf{0.876 \pm .004}$ | $\mathbf{0.876 \pm .004}$ |
| credit-g | $0.767 \pm .042$ | $0.762 \pm .043$ | $\mathbf{0.770 \pm .034}$ | $\mathbf{0.775 \pm .040}$ | $0.769 \pm .044$ | $\mathbf{0.773 \pm .039}$ |
| eucalyptus | $0.780 \pm .035$ | $\mathbf{0.780 \pm .032}$ | $0.778 \pm .034$ | $\mathbf{0.833 \pm .000}$ | $0.779 \pm .037$ | $0.780 \pm .036$ |
| first-order-theorem-proving | $0.824 \pm .012$ | $\mathbf{0.824 \pm .012}$ | $\mathbf{0.825 \pm .012}$ | $0.820 \pm .017$ | $\mathbf{0.824 \pm .013}$ | $\mathbf{0.825 \pm .011}$ |
| gesturephasesegmentationprocessed | $0.888 \pm .009$ | $\mathbf{0.892 \pm .011}$ | $0.863 \pm .027$ | $0.781 \pm .055$ | $0.874 \pm .044$ | $0.887 \pm .017$ |
| jannis | $0.851 \pm .004$ | $\mathbf{0.856 \pm .004}$ | $0.843 \pm .014$ | $0.790 \pm .068$ | $\mathbf{0.851 \pm .004}$ | $0.846 \pm .017$ |
| kc1 | $0.789 \pm .039$ | $\mathbf{0.798 \pm .041}$ | $\mathbf{0.791 \pm .038}$ | $\mathbf{0.792 \pm .039}$ | $0.790 \pm .040$ | $\mathbf{0.791 \pm .034}$ |
| kick | $0.770 \pm .009$ | $\mathbf{0.771 \pm .008}$ | $0.770 \pm .009$ | $0.770 \pm .010$ | $\mathbf{0.771 \pm .008}$ | $\mathbf{0.771 \pm .009}$ |
| kr-vs-kp | $1.000 \pm .000$ | $1.000 \pm .000$ | $\mathbf{1.000 \pm .000}$ | $1.000 \pm .000$ | $\mathbf{1.000 \pm .000}$ | $1.000 \pm .000$ |
| numerai28.6 | $0.523 \pm .003$ | $\mathbf{0.523 \pm .003}$ | $0.519 \pm .006$ | $0.509 \pm .010$ | $0.520 \pm .006$ | $0.522 \pm .003$ |
| okcupid-stem | $0.839 \pm .003$ | $\mathbf{0.845 \pm .004}$ | $0.844 \pm .005$ | $0.844 \pm .005$ | $0.841 \pm .007$ | $\mathbf{0.845 \pm .005}$ |
| phishingwebsites | $0.996 \pm .001$ | $\mathbf{0.997 \pm .001}$ | $0.996 \pm .001$ | $0.996 \pm .001$ | $\mathbf{0.996 \pm .001}$ | $0.996 \pm .001$ |
| phnome | $0.956 \pm .009$ | $\mathbf{0.959 \pm .011}$ | $0.944 \pm .017$ | $0.890 \pm .039$ | $0.954 \pm .016$ | $0.954 \pm .015$ |
| qsar-biodeg | $0.925 \pm .044$ | $\mathbf{0.929 \pm .040}$ | $0.925 \pm .042$ | $0.923 \pm .041$ | $\mathbf{0.926 \pm .042}$ | $\mathbf{0.926 \pm .044}$ |
| satellite | $0.987 \pm .014$ | $\mathbf{0.992 \pm .008}$ | $0.990 \pm .010$ | $0.988 \pm .014$ | $0.985 \pm .024$ | $\mathbf{0.988 \pm .014}$ |
| segment | $0.996 \pm .002$ | $0.996 \pm .002$ | $0.996 \pm .002$ | $0.991 \pm .003$ | $0.996 \pm .002$ | $0.996 \pm .002$ |
| shuttle | $0.589 \pm .054$ | $\mathbf{0.646 \pm .060}$ | $0.602 \pm .061$ | $\mathbf{0.605 \pm .054}$ | $\mathbf{0.629 \pm .057}$ | $\mathbf{0.614 \pm .063}$ |
| sylvine | $0.986 \pm .004$ | $\mathbf{0.993 \pm .003}$ | $0.984 \pm .006$ | $0.968 \pm .01$ | $\mathbf{0.986 \pm .004}$ | $\mathbf{0.986 \pm .004}$ |
| vehicle | $0.933 \pm .013$ | $\mathbf{0.942 \pm .018}$ | $0.932 \pm .016$ | $0.925 \pm .022$ | $0.932 \pm .018$ | $0.932 \pm .014$ |
| wilt | $0.990 \pm .013$ | $\mathbf{0.994 \pm .005}$ | $\mathbf{0.990 \pm .012}$ | $0.990 \pm .011$ | $\mathbf{0.992 \pm .008}$ | $\mathbf{0.992 \pm .010}$ |

