# OpenReview forum: "Large Language Models Engineer Too Many Simple Features for Tabular Data"
_ICLR.cc/2025/Conference — ICLR 2025 Conference Withdrawn Submission_

### Official Review · Reviewer_31oL · 2024-11-02

**Soundness:** 2
**Presentation:** 2
**Contribution:** 1
**Rating:** 3
**Confidence:** 4

**Summary:**

This paper investigates Large Language Models' (LLMs) capabilities in feature engineering for tabular data. The study examines four LLMs across 27 tabular classification datasets that were specifically selected to minimize potential memorization effects. For each dataset, the LLMs were tasked with recursively generating 20 new features, using prompts that contained task context, descriptions of original features, and available operators. The study benchmarks these results against OpenFE, an open-source feature engineering tool, using identical operators and original features. To evaluate the effectiveness of the engineered features, a LightGBM model was trained and tested on datasets combining original and constructed features, with classification accuracy as the performance metric. The results demonstrate that OpenFE produces more consistently beneficial feature sets for classification tasks. Through analyzing operator frequency in feature construction across both LLMs and OpenFE, the authors conclude that LLMs exhibit a bias toward simpler operators when no explicit operator preferences are specified in the prompts.

**Strengths:**

The paper presents a novel investigation into LLMs' feature engineering capabilities. The authors introduce an innovative evaluation metric—operator frequency distribution—which effectively quantifies the patterns in operator selection during feature construction. This metric provides valuable insights into how feature engineering tools, particularly LLMs, exhibit preferences for certain operators under different task contexts and prompt conditions. Furthermore, the study's comprehensive evaluation across 27 tabular datasets, with careful consideration for LLM memorization effects, demonstrates robust experimental design and systematic methodology.

**Weaknesses:**

The paper's analysis lacks sufficient depth in several crucial areas. While the proposed operator frequency metric is interesting, it requires further validation in terms of:

Effectiveness: There is no analysis comparing the variability and information content of features generated by simple versus complex operators.
Fairness: The operator-level analysis overlooks that identical operators applied to different features can yield vastly different outcomes, making tool comparisons based solely on operator frequency potentially misleading.
Implications: The study lacks experimental evidence linking complex operator usage to improved classification performance.

The paper's conclusion about LLMs' preference for basic operators requires additional validation. The authors did not explore prompting strategies to encourage complex operator usage, nor did they analyze the specific features and operators suggested by LLMs.
The narrative structure could be improved. For instance, the abstract's discussion of LLM bias in text generation appears tangential to the core focus on feature engineering. Similarly, the section on 'Other Applications of Large Language Models for Tabular Data' would be better integrated into the literature review rather than appearing as a standalone paragraph.

**Questions:**

1, Could you clarify the source of the original features—were they extracted or provided with the datasets?
2, Have you considered experimenting with prompts that encourage the use of complex features, perhaps by emphasizing intricate relationships between original features?
3, What methods were used to validate the effectiveness, fairness, and implications of the operator frequency metric?
4, How did you account for the stochastic nature of LLM responses, where identical prompts might yield different operators and features?
5, Would it also be informative to evaluate model performance using only the generated features, excluding original features? Maybe you can try this.
6, Have you conducted feature-level analysis of the constructed features? Specifically:
Classification Performance Level: Identifying dominant features in both LLM and OpenFE-generated sets
Feature Level: Analyzing the characteristics of successful versus unsuccessful generated features
Combining classification-level, feature-level, and operator-level analyses to strengthen conclusions about LLMs' feature engineering capabilities.
7, A potential typo in Hypothesis 1: “HYPOTHESIS 1: FEATURE ENGINEERING WITH LARGE LANGUAGE MODELS IS BIASED TOWARD SIMPLE OPERATES.” The last word should be “OPERATORS”?

---

> ### Author Response · Authors · 2024-11-27
> **Response Reviewer 31oL [1/2]**
>
> Dear Reviewer 31oL,
>
> Thank you for reviewing our paper. We are excited to see that you see our evaluation pipeline's robustness and systematic nature. We also noticed your critiques and will address them below.
>
> In summary, your review focuses on the alleged lack of depth, inadequate validation of the operator frequency metric, unsubstantiated conclusions, and a narrative structure that detracts from its overall coherence. We hope that we understand your concerns correctly. If not, we encourage you to reiterate your concerns and explain them to us in greater detail.
>
> Assuming we understand your concerns correctly, we will address your weaknesses.
>
> ## Weaknesses
> > Effectiveness: There is no analysis comparing the variability and information content of features generated by simple versus complex operators
>
>
> We consider an in-depth analysis of features generated by simple vs. complex operators out of scope for this work. The scope of our work is to detect a bias. As we show, it was sufficient to compare operators used by existing automated feature engineering methods with context-aware feature engineering methods (using LLMs). We do not see how characteristics of the features would support our claim, especially as this information is conditional on the datasets and not comparable between multiple datasets.
>
> From comparing the operator distributions, we were able to see a clear disconnect between the operators selected by OpenFE and the ones selected by the LLMs. This leads to the second hypothesis of this paper, which is that this bias negatively impacts the value of features (in a general case). This was apparent through comparing predictive accuracy results where LLM-based methods were constantly subpar to non-LLM-methods (OpenFE).
>
> > Fairness: The operator-level analysis overlooks that identical operators applied to different features can yield vastly different outcomes, making tool comparisons based solely on operator frequency potentially misleading
>
> We ask you top clarify how the described problem is a fairness problem or potentially misleading.
>
> Our conclusion is not affected by which features are selected, because the bias exists no matter the features (the prompt does not change based on the select feature) and the performance is a suitable indicator for the impact across many datasets. Moreover, we show that the feature selection part of our pipeline has no significant impact, see Appendix G.3.
>
> > Implications: The study lacks experimental evidence linking complex operator usage to improved classification performance.
>
> We do not claim anywhere in the paper that complex operators lead to improved classification performance. This would also be a false implication, as complex operators do not necessarily lead to better performance in all cases. We solely claim that when considering the distribution of operators of OpenFE (which has a notably higher usage of complex operators) and comparing it with those of an LLM-based method (which in some cases uses almost no complex operators at all), the outcome in predictive accuracy seems to be on average superior when spreading better across all types of operators. In other words, we claim that there are situations in which a complex operator may be useful, which the LLM fails to realize as it almost never suggests complex operators as useful transformations.
>
> > The authors did not explore prompting strategies to encourage complex operator usage, nor did they analyze the specific features and operators suggested by LLMs.
>
> We considered this point and added Figure 11 in Appendix G.3, comparing which features the LLM selected. From this figure, we conclude that the LLM is certain which features it considers important for feature engineering in most cases; however, it fails to try different combinations of these features with different operators.
>
> Our work explicitly calls for this as future work; we further clarified this in the updated manuscript. Moreover, we unsuccessfully tried different prompting strategies to avoid bias in our own applications (as detailed in another response to a review above).

---

> ### Author Response · Authors · 2024-11-27
> **Response Reviewer 31oL [1/2]**
>
> > The narrative structure could be improved. For instance, the abstract's discussion of LLM bias in text generation appears tangential to the core focus on feature engineering. Similarly, the section on 'Other Applications of Large Language Models for Tabular Data' would be better integrated into the literature review rather than appearing as a standalone paragraph.
>
> These are two points on which we do not understand your concerns. In a paper where the main focus of the work is a bias in LLMs, we consider biases in LLMs for text generation important related work as it lays a foundational understanding of bias in LLMs. For the second part, it is obvious from the paper that the paragraph “Other Applications of Large Language Models for Tabular Data” is part of the related work. Please reiterate what you mean by “[...] would be better integrated into the literature review [...]” and how the narrative structure, specifically in the context of our work, can be improved.
>
> # Questions
> 1. We do not entirely understand what you mean by this question. For benchmarking, we used well-known tabular datasets from the AutoML benchmark [1], which consist of features and data points. So, the original features are already part of the dataset.
> 2. No, we have not done this yet. This is mainly due to two factors. First, we wanted an unbiased evaluation where every operator is weighted equally so we could see the true distribution of the LLMs. Second, we expect the LLM to know the correct operators in fitting cases. As stated above, complex operators may not always be a useful solution. However, we want the LLM to know when to select complex operators and when not.
> 3. Refer to the Weakness section, where we explain our opinion on all of the stated problems.
> 4. Unfortunately we do not entirely understand what you mean by this question. We don't expect the LLM to be deterministic at all (e.g. always yield the same solution for an example dataset incorporated into our prompt). We just want to evaluate whether the LLM is able to also propose complex operators. This is also why we repeated our experiments across many datasets and repeated prompts per dataset.
> 5. In our opinion, this would not be useful as we are not interested in finding the optimal feature combination when predicting on a given task using solely this newly generated feature, nor do we expect the LLM to find this. What we consider here is a data science expert wanting to employ an LLM for automated feature engineering on a given tabular data problem, a setting present in well-known methods like CAAFE [2], DS-Agent[3] or FELIX[4]. In this case, the expert would expect the LLM to find a fitting solution for the given table (which the expert would probably not drop entirely as this would lose information).
> 6. As shown in the plots in our results, we have conducted operator level analysis (Figure 2, Figure 3, Figure 4), and feature level analysis (Figure 11), as well as classification level analysis (Figure 5, Table 3)
> 7. Thank you! We have fixed this in our revised paper version.
>
> [1] Gijsbers, Pieter, et al. "An open source AutoML benchmark." Journal of Machine Learning Research 25.101 (2024), https://arxiv.org/abs/2207.12560
> [2] Hollmann et al. “Large Language Models for Automated Data Science: Introducing CAAFE for Context-Aware Automated Feature Engineering”, Conference on Neural Information Processing Systems (2023), https://arxiv.org/abs/2305.03403
> [3] Guo et al. “DS-Agent: Automated Data Science by Empowering Large Language Models with Case-Based Reasoning”, ICML (2024), https://arxiv.org/abs/2402.17453
> [4] Malberg et al., “FELIX: Automatic and Interpretable Feature Engineering Using LLMs”, Joint European Conference on Machine Learning and Knowledge Discovery in Databases (2024), https://link.springer.com/chapter/10.1007/978-3-031-70359-1_14

---

### Official Review · Reviewer_bAV4 · 2024-11-02

**Soundness:** 4
**Presentation:** 3
**Contribution:** 2
**Rating:** 5
**Confidence:** 4

**Summary:**

The paper tests LLM bias in the task of feature engineering for training a downstream learning system. The paper uses several LLM settings across 27 datasets, to demonstrate that LLMs do indeed have bias for this task, which indeed seems to lead to poor performance compared to an existing SOTA automated feature engineering solution. Some further discussion and experimentation into the properties of the bias shows that the LLM seems to prefer simpler features when the features have meaningful names, but also doesn't sample uniformly when the features have nondescript random names.

**Strengths:**

The paper is based on solid experimental work, testing using several LLMs and across many datasets, testing for memorization issues separately to check for bias explicitly.
The paper is an interesting and easy to follow read. Problematic properties of LLM solution paths for different problems are always appreciated, as we develop more and more systems that significantly rely on this tool, we must strive to understand the biases this seemingly easy fix-all solution of asking an LLM brings into our work and the times it might fail completely. It is also interesting that the large models have failed worse at adding features that helped with the downstream system's results compared to the smaller models, which did help a little.

**Weaknesses:**

The main issue with this paper is that it is rather unclear why the usage of LLMs for this task was explored at all. It seems that when feature engineering is done by an LLM, the downstream system's performance is worse than existing SOTA systems -  and sometimes even worse than doing any feature engineering at all. Frankly, it's also not a task that I would intuitively expect LLMs to be good at, as general knowledge, common sense and language knowledge is probably not what humans would use for feature engineering, but rather math/engineering skills and perhaps expert domain knowledge or lived experience - all usually less strong qualities of LLMs. The paper does not call this issue out or justify it. Usually, checking for biases of a solution might have one of two purposes: 1. call out poor performance that happens in a way that isn't expected or measured in other ways, so for example, if the system had seemingly good downstream performance, checking for biases or other issues might help guard us from using a problematic solution that looks good in our metrics. 2. try to improve the performance of the biased system by somehow mitigating the bias. It seems that option 1 in this case is unnecessary, since the LLMs have worse performance, and no actual attempt is made towards the #2 target.

**Questions:**

1. Why would I use an LLM for feature engineering anyway, if there are existing SOTA automated systems that already do it and perform much better?
2. If your answer to #1 is that I probably wouldn't, then the main question about publishing this paper would be - why would I read a paper about the biases such a solution might have? There could be several answers (e.g., to inspire a similar analysis of LLM solutions for other problems) but they need to be clear within the paper.

Small issues:
1. Please explain that the improvement in "Predictive Performance Improvement" is improvement compared to a system without FE earlier in the document, e.g. before table 3.
2. While the random experiment is fun and adds to the paper, I don't think it is at all accurate to say that it tests "whether our prompt template influenced our results" - seeing as the prompt template itself did not change in this experiment, only the names of the features. I don't think it shows anything about the prompting strategy - but rather that the nature of the bias depends on the feature naming.
3. Caught typo: The API usage costed -> cost

---

> ### Author Response · Authors · 2024-11-18
>
> Dear Reviewer bAV4,
>
> We highly appreciate the effort taken to review our paper and thank you for the valuable feedback. We are happy to see your satisfaction with our general experimental workflow. Additionally, we thank you for highlighting one of our takeaways - that one has to be careful when using an LLM as a fix-all solution - as they are possibly limited by negative biases. We carefully considered the concerns you raised in your review and aim to address them below
>
> ## Weaknesses
>
> > *The main issue with this paper is that it is rather unclear why the usage of LLMs for this task was explored at all*
>
> We base our work on the existence of different applications of LLMs in regard to feature engineering for tabular data. We reviewed and presented them in the “Related Work” section of our paper, where we included a paragraph on “Feature Engineering with Large Language Models”. Apart from multiple different approaches in this area of work, our work was mainly motivated by the paper “CAAFE - Context Aware Automated Feature Engineering” which introduced a novel approach to using LLMs to generate new features on a tabular dataset. Driven by the success of this method (successful improvements in predictive accuracy on 12/14 benchmark datasets, results from CAAFE paper), our own investigation led us to the bias we discovered in LLMs when employing them for such a task, a problem which in our opinion can’t be neglected when using LLMs for feature engineering. Moreover, a problem that follow-up work for CAAFE seems to be ignorant of so far.
>
> We added the reference from related work to the introduction to clarify the relationship and motivation.
>
> > *Frankly, it's also not a task that I would intuitively expect LLMs to be good at, …, expert domain knowledge or lived experience, …*
>
> We generally agree with you about LLMs' capabilities in that regard. However, all existing related work (and papers submitted to ICLR) would disagree with us and have applied LLMs to this task. We note that most of them also do not perform memorization tests because they want to show their method is better while we try to really understand the limitation (resulting from the bias) for the downstream tasks of LLMs.
>
> Likewise, our primary motivation is not to expect the LLM to act as a highly knowledgeable expert on any given topic but rather to exploit some capabilities of understanding semantic context information with a given dataset, a set of information usually available for tabular datasets. Existing black-box feature engineering methods do not use such context information, which motivated most related work. The general motivation is therefore that this information can carry weight, but it is currently not used in most methods.
>
> > *and no actual attempt is made towards the #2 target*
>
> Our work is a call to action for the community, related work, or concurrent work. Likewise, it functions as a warning to practitioners or Kaggle experts who are actively using or considering using LLMs for their applications.
>
> We believe our contribution highlights the unexpected existence of a bias that prompts poor performance, which is valuable research. Moreover, it is research that proposes a method to detect such a bias that requires scrutiny of peer review. Otherwise, practitioners and related or concurrent work will never consider investigating such a problem in their own systems.
>
> In conclusion, our work provides a meaningful contribution even though we did not address #2 in the paper.
> Nevertheless, we would like to mention that we tried to address #2 in our own systems and applications but have not found a solution that avoids bias (via in-context learning or fine-tuning of an LLM). This, however, does not mean there are no possible solutions. Thus, we did not believe that such a negative result as part of the appendix would contribute anything meaningful to the goal of this paper.

---

> ### Author Response · Authors · 2024-11-18
>
> ## Questions
> 1. Refer to the explanation above, but in short, we consider semantic information an important addition to tabular datasets. Existing SOTA methods are incapable of incorporating this information. LLMs are currently the best alternative we have at hand to process semantic information and gain some form of reasoning capabilities, which can be valuable when considering a tabular dataset in light of its semantic description.
> 2. Refer to the explanation above, but in short: methods that employ LLMs for the given task exist in multiple forms and by different users, so there is a real application of this right now. Therefore, an existing bias that might mitigate the capabilities of these approaches is essential information to be made aware of when working on/with such methods.
>
> ## Small Issues
> 1. Thank you for highlighting this, we have improved the explanation in the paper. (L353-354)
> 2. Regarding the random experiment: Our primary motivation was to strengthen the validity of our prompting approach further. We also masked the names of the operators in this experiment. The idea behind this experiment was to mitigate the possibility of this bias not because the LLM is biased towards some operators but because it just always selects the first operator that comes in the list of possible operators in the prompting template. That is why we masked the names to solidify that the LLM can pick operators from every position in the prompt and that our experienced bias is not just a (strong) positional bias. We also agree that the statement was too strong, so we have toned it down and made it more specific to what the experiment shows. (Refer to Lines 434-435 and 448-449)
> 3. Thanks!
>
> We sincerely hope that this response helps in answering your questions and concerns. If you have any more questions or concerns regarding our work, we are happy to answer them as well.
>
> Furthermore, we thank you for the insightful discussion and for giving us the opportunity to clarify and defend our contribution. We would be grateful to hear any further comments about the motivation of our work.
>
> Thank you very much, and kind regards,
> The Authors

---

> ### Author Response · Authors · 2024-11-25
>
> Dear Reviewer bAV4,
>
> We would again like to thank you for your insightful comments regarding our submission. We hope that our revision resolved your questions and concerns. We would like to kindly ask, if our revisions influenced your score of our submission. If any further clarification is needed, please let us know.
>
> Thank you, and kind regards,
> The Authors

---

### Official Review · Reviewer_oDSt · 2024-11-04

**Soundness:** 3
**Presentation:** 3
**Contribution:** 1
**Rating:** 3
**Confidence:** 5

**Summary:**

The paper explores the featuring engineering powers of LLM with OpenFE as the baseline. The authors perform experiments on 27 datasets and 4 LLMs. The primary findings are the following.

1. LLM perform worse than the baseline.
2. Proprietary "Large" models perform worse than small "open" models.

**Strengths:**

The authors experimentally show the limitations of LLMs for feature engineering. The experimental setting is convincing.

**Weaknesses:**

1. The conclusions of the paper are along expected lines and are not surprising. A more notable contribution would be to address the limitations.
2. The statistical significance of the results is not provided.
3. The term "bias" is too strong for the problem explored. The authors can use the word "limitation".

**Questions:**

1. What is the statistical significance of the results shown in Table 3?
2. Why aren't larger models in GPT and Gemini family not explored?

---

> ### Author Response · Authors · 2024-11-18
>
> Dear Reviewer oDSt,
>
> Thank you for your time and effort in reviewing our paper. We are happy to see that you find our evaluation setup convincing and that you additionally see our contribution in showing the limitations of LLMs in regard to feature engineering. We also appreciate your valuable critiques, and in an effort to address your concerns, we carefully considered them and aim to address them below.
>
> In your summary, you focus on the predictive performance of the LLM. Similarly, in the weaknesses and questions, you also focus on the predictive performance of the method / LLMs. Our work does not focus on predictive performance. We focus on the bias of the generated output, which, coincidently, results in worse predictive performance. This bias could also exist without a drop in performance (e.g., the LLMs could be good because they engineer too many simple features).
>
> Therefore, we would like to know if you have other concerns about our method that determines the bias that prompted your negative assessment of our contribution and work. Specifically, do you see a problem with our analysis of the bias of LLMs?
>
> ## Weaknesses
> 1. This is not an expected result for us, especially considering prior work and practitioners actively using LLMs for feature engineering. We would be grateful if you could point us to related works that show similar conclusions to those our work has found.
> To illustrate, consider the following: there are possible features that can be transformed into new valuable features. These features can be transformed using complex transformations, which are not easy to find except if you understand the context of the data. Now, if an LLM had no contextual understanding of the data (similar to traditional feature engineering methods which do not employ LLMs for context understanding), we would expect this LLM to perform something similar to a random search over all features and operators, as every operator should be similarly weighted at first. However, this is not the case, and the LLM heavily relies on few (simple) operators; without these operators resulting in strong improvement in predictive accuracy.
> This is unexpected, especially since our community (e.g., prior work and Kaggle) believes that LLMs are capable of good feature engineering [1], [2], [3].
>
> 2. Thank you for pointing this out. We have now added an evaluation of statistical significance for the predictive performance using a CD diagram [4] in Figure 9. It clarifies that OpenFE, considering its rank, is superior to all LLM-based methods. Moreover, OpenFE is significantly different from Gemini and GPT-4o-mini. Additionally, it shows the similarities between the capabilities of GPT-4o-mini and Gemini-1.5-flash as well as the slightly better performance of Llama3.1-8b and Mistral-7b-v0.3 which are again similar to each other, as also stated in the paper (Table 3, Figure 5).
>
> 3. Relating to the above explanation, we would like to stand by the word bias. An unbiased LLM could perform similarly to a random search. It would, therefore, result in a smoother distribution over all operators, which is not the case. The term bias, in this sense, is, of course, only related to the affinity of an LLM towards proposing certain mathematical operators. Furthermore, the word bias has a very clear meaning in the related work and literature related to analyzing LLMs [5].

---

> ### Author Response · Authors · 2024-11-18
>
> ## Questions
> 1. See Figure 9.
> 2. This was mainly because we wanted to use publicly available API endpoints and LLM providers, as most practitioners usually do. As a large-scale evaluation was important to us to have a strong foundation for our results, the API costs for more powerful models would not be feasible for our evaluation. However, we updated our submission to provide Appendix G.1 Figure 10, where we benchmarked our idea on a subset of datasets and compared the distributions of operators between GPT-4o-mini and the more powerful GPT-4o. As apparent in this figure, the bias is similar, and especially for the GPT models, the fact that they have a very limited range of operators they select at all and still do not select complex operators remains the same.
>
> We sincerely hope that this response helps in answering your questions and concerns. If you have any more questions or concerns regarding our work, we are happy to answer them as well.
>
> Furthermore, we would be grateful for your opinion on our core contribution, the method for determining bias, and how we could improve it.
>
> Thank you very much, and kind regards,
> The Authors
>
> ## References
> [1] Hollmann et al. “Large Language Models for Automated Data Science: Introducing CAAFE for Context-Aware Automated Feature Engineering”, Conference on Neural Information Processing Systems (2023), https://arxiv.org/abs/2305.03403
> [2] Han et al. “Large Language Models Can Automatically Engineer Features for Few-Shot Tabular Learning” (2024), https://arxiv.org/abs/2404.09491
> [3] Zhang et al, “ELF-Gym: Evaluating Large Language Models Generated Features for Tabular Prediction”, Proceedings of the 33rd ACM International Conference on Information and Knowledge Management, (2024), https://dl.acm.org/doi/abs/10.1145/3627673.3679153
> [4] Demšar, Janez. "Statistical comparisons of classifiers over multiple data sets." The Journal of Machine learning research 7 (2006): 1-30.
> [5] Navigli et al, “Biases in large language models: origins, inventory, and discussion” ACM Journal of Data and Information Quality (2023), https://dl.acm.org/doi/10.1145/3597307

---

> > ### Comment · Reviewer_oDSt · 2024-11-27
> > **Thanks for the response**
> >
> > I thank the authors for responding to my comments, especially, for providing statistical significance results and providing additional references.
> >
> > I looked at the paper “Biases in large language models: origins, inventory, and discussion” and the primary sense of the word "bias" in the paper is "social bias". This was my point.
> >
> > The authors have asked "We would be grateful if you could point us to related works that show similar conclusions to those our work has found."
> >
> > I will encourage the authors to look at McCoy, R. T., Yao, S., Friedman, D., Hardy, M., & Griffiths, T. L. (2023). Embers of autoregression: Understanding large language models through the problem they are trained to solve. This paper shows the many of the surprising behaviors or hallucinations or "biases" of LLMs can be traced back to the training data. My hypothesis is that simple operators abound in web data and this is mirrored in the output. I do not have any other paper in mind.
> >
> > Having read the thoughtful response and being appreciative of it, I am keeping the scores the same.

---

> > > ### Author Response · Authors · 2024-11-27
> > > **RE: Dispute Over the Use of the Word Bias**
> > >
> > > Dear Reviewer oDSt,
> > >
> > > We would like to reiterate your comments about bias, as there seems to be a significant disconnect between our opinions. It appears you are rejecting our work solely because we called the observed phenomenon a bias, even though we addressed all your other concerns.
> > >
> > > **We strongly believe that bias is the correct formulation to use in our work, following from its definition, the related work you referenced, and the context of our work.**
> > >
> > > The definition of bias is [1]:
> > > > “A tendency, inclination, or leaning towards a particular characteristic, behaviour, etc.;”, “Distortion of a statistical result arising from the method of sampling, measurement, analysis, etc.;”
> > >
> > > The first definition strongly matches our work as we have found a clear tendency of LLMs to lean towards a particular characteristic (i.e., simple operators).
> > >
> > > Additionally, from your referenced paper by McCoy et al. (Section 3.3):
> > > > “Specifically, we predict that LLMs will be biased toward producing high-probability sequences of words, meaning that their performance will get worse when the correct output is in fact low-probability.”
> > >
> > > This matches our work and our general assumption that the usage of complex operators is just too sparsely represented in training data, as you also stated in your response, which leads to the LLM rarely selecting these operators, even though they can be useful in many cases. **Note that the work you referenced specifically names this a “bias”** for exactly these situations where the LLM is biased because of low probabilities in the output sequence.
> > >
> > > The origin of the bias does not change the fact that it is still a bias. A model that was trained on biased data (e.g., in fairness problems) is still considered biased.
> > >
> > > Lastly, as mentioned in our related work section where we consider “Biases in LLMs”, we are aware of the traditional usage of the word bias in a social context and clearly stated that we are looking at something similar but different (Line 98).
> > >
> > >
> > >
> > > Kind Regards,
> > >
> > > The Authors
> > >
> > > ---
> > > [1] http://www.oxforddictionaries.com

---

> ### Author Response · Authors · 2024-11-25
>
> Dear Reviewer oDSt,
>
> We deeply hope that our response addressed all of your questions about our submission. We would therefore like to kindly ask if you reconsidered your score.
>
> Kind Regards,
> The Authors

---

### Official Review · Reviewer_JGk8 · 2024-11-04

**Soundness:** 2
**Presentation:** 2
**Contribution:** 2
**Rating:** 3
**Confidence:** 4

**Summary:**

The authors investigate how well LLMs can engineer features for tabular datasets. Specifically, they look at the frequencies of operators, and find that there is bias toward simpler features rather than more interesting or useful ones. They also evaluate on the downstream accuracy of the models trained with and without the engineered features.

**Strengths:**

It is good to see more examples of evaluation of downstream LLM tasks "in the wild".

I appreciate that the authors were rigorous in removing datasets that were thought to be memorized or in the training data of the LLM, even though they did not have access to the training data itself.

**Weaknesses:**

To me, this doesn’t seem like an influential enough contribution. Not only is it tackling a very narrow problem, but it is also only evaluating a specific method for addressing that problem. While there is some prior work around using LLMs for feature engineering, I’m not convinced that this work’s method for feature engineering is necessarily representative of all the methods for using LLMs for this task.

Specifically, the authors only use one prompting strategy, on a snapshot of models at the current time that this paper is being written. A few examples of people using LLMs for feature engineering are cited (Hatch, 2024; Türkmen, 2024), but it is unclear what methods these citations used– is the author’s method the same, or inspired by them? Should data scientists conclude from this paper that they should never use LLMs for feature engineering, even if they use a different method? Overall, I think this is an interesting use case to evaluate, but the work is not broad enough to be included in ICLR.

Nits:
Typo: “which is send to the LLM” → sent

**Questions:**

I’m a little confused about the experimental setup regarding operations. If I understand correctly, the authors are comparing the distribution of operators generated by the LLM and by OpenFE. If OpenFE is considered ground truth, why not compare directly to OpenFE final generated feature set? For example, rather than just counting the number of times we see the operation “GroupByThanRank”, why not look at the original features that were input into this operation?

---

> ### Author Response · Authors · 2024-11-18
>
> Dear Reviewer JGk8,
>
> We highly appreciate your effort to review our paper, and thank you for the valuable feedback. We thank you for your positive feedback on the precision of our work, specifically highlighting our action of memorization tests before evaluating LLMs.
>
> We carefully considered the concerns you raised in your review and aim to address them below.
>
> ## Weaknesses
>
> > “ tackling a very narrow problem”
>
> Feature engineering is a highly important problem for tabular data. Tabular data problems are omnipresent in real-world applications, especially in industry. Thus, while we agree that this is a narrow problem from the point of view of the LLM space, our work tackles a major problem in the world of tabular data—which is the focus of our work. Improving LLMs specifically for downstream tasks related to tabular data can have major positive implications for many applications.
>
> > “, but it is also only evaluating a specific method for addressing that problem”
>
> As described in Section 3 Stage B), the method we created was chosen primarily to enable our analysis (e.g., by avoiding code generation failure). Our work tries to analyze the world knowledge of LLMs and not a specific method for feature engineering. Our method enables such an analysis, and we would appreciate specific feedback as to why it would not be representative of the world knowledge of an LLM.
>
> > “..., on a snapshot of models”
>
> We used some of the most well-known models. OpenAI's GPT models are especially heavily used in most work with LLMs as a benchmark, specifically in related work. Data Science tasks with LLMs usually use GPT4 and Llama models as a benchmark. Hence, the evidence that these models induce such a strong bias can be valuable information for fellow researchers. Furthermore, we added an additional experiment with GPT4o in Figure 10, one of the strongest models on the market (according to LLM arena [1]), which also exhibits the bias presented in our work. We were not able to use Claude due to rate limiting of the API.
>
> > *A few examples of people using LLMs for feature engineering are cited (Hatch, 2024; Türkmen, 2024), but it is unclear what methods these citations used*
> We specifically refer to these authors as both rely on CAAFE [2] (which we also refer to in the related work section) for well-known Kaggle competitions. Both these citations strengthen our motivation that automated feature engineering methods with LLMs are used in practice. In academia, we can see CAAFE being used or built upon in several related works, such as DS-Agent [3], FELIX [4], or ELF-Gym[5].  In short, CAAFE is, to the best of our knowledge, the most prominent feature engineering method with LLMs.
>
> CAAFE is especially relevant to us as this method also follows an approach where the LLM generates new features from existing ones by applying operators on these existing features. The core approach is highly similar, but to enable our analysis, we decided to let the LLM provide us directly with the operators it would like to apply rather than parsing the generated Python code (CAAFE’s method of feature generation).
>
> We adjusted the paper to reflect this connection better and highlight CAAFE in greater detail; refer to lines 89-93.
>
> > *Should data scientists conclude from this paper that they should never use LLMs for feature engineering, even if they use a different method?*
>
> In our opinion, data scientists should conclude from this method that LLMs can be employed for feature engineering; however, when employing the LLM to directly generate new features from existing ones with mathematical operators (like CAAFE, which is used in practice as followed from the above talked about citations), one has to consider that this is biased and the LLM might fail to find more complex solutions as it heavily relies on simple operators. Therefore, our work shows the tabular community that new research efforts to further strengthen LLMs are needed, genuinely enabling the use of LLMs for tabular downstream tasks. Likewise, it shows the LLM community that we require methods to explore and avoid bias when using LLMs for downstream tasks.
>
> We now further clarify this call for action and takeaway in our conclusion.
>
> > “Nits: Typo: “which is send to the LLM” → sent”
> Thank you for pointing this out; we fixed it in the updated paper version. Ref: L:732

---

> ### Author Response · Authors · 2024-11-18
>
> ## Questions
>
> > “If OpenFE is considered ground truth, why not compare directly to OpenFE final generated feature set?”
>
> Similar to previous work on text generation [6], we are looking for trends in the distribution of the output of an LLM to obtain a representative, meaningful conclusion.
>
> A direct “match” comparison to OpenFE’s feature set would fail to capture this. For example, if the LLM suggested using different but still complex operators such as GroupByThenMean instead of OpenFE’s GroupByThenRank, that would not prompt a negative bias. Yet the LLM would have “failed” in the direct comparison to OpenFE.
>
> > *why not look at the original features that were input into this operation?*
>
> We have added an additional figure for that in Appendix G.2 Figure 11. Here, we present the frequencies with which the LLM selects each feature from each dataset. As apparent, in most cases, the LLM is reasonably sure which features it would like to transform, as indicated by the relatively high frequencies for certain features. However, it is then, in our opinion, limited by the fact that it repeatedly uses the same simple operators on these features instead of trying different operators on the same problem (an approach similar to random search, which the method would suggest one would do if one had 1: no understanding of the underlying context, 2: an unbiased opinion on all operators). This further strengthens our conclusion that this is a bias problem regarding the existing operators.
>
> We sincerely hope that we were able to clarify your questions and concerns. If you have any more questions or concerns regarding our work, we are happy to answer them as well.
>
> Furthermore, we would be grateful for any specific pointers to improve our work further.
>
> Thank you very much, and kind regards,
> The Authors
>
> ## References:
> [1] Chatbot Arena, https://lmarena.ai/
> [2] Hollmann et al. “Large Language Models for Automated Data Science: Introducing CAAFE for Context-Aware Automated Feature Engineering”, Conference on Neural Information Processing Systems (2023), https://arxiv.org/abs/2305.03403
> [3] Guo et al. “DS-Agent: Automated Data Science by Empowering Large Language Models with Case-Based Reasoning”, ICML (2024), https://arxiv.org/abs/2402.17453
> [4] Malberg et al., “FELIX: Automatic and Interpretable Feature Engineering Using LLMs”, Joint European Conference on Machine Learning and Knowledge Discovery in Databases (2024), https://link.springer.com/chapter/10.1007/978-3-031-70359-1_14
> [5] Zhang et al, “ELF-Gym: Evaluating Large Language Models Generated Features for Tabular Prediction”, Proceedings of the 33rd ACM International Conference on Information and Knowledge Management, (2024), https://dl.acm.org/doi/abs/10.1145/3627673.3679153
> [6] Liang et al., “Monitoring AI-Modified Content at Scale: A Case Study on the Impact of ChatGPT on AI Conference Peer Reviews” (2024),
> https://arxiv.org/abs/2403.07183

---

> ### Author Response · Authors · 2024-11-25
>
> Dear Reviewer JGk8,
>
> We hope that our responses were able to address your concerns and answer your questions regarding our submission. We would kindly ask if they led you to reconsider your score. Please let us know if further clarifications are needed.
>
> Kind Regards,
> The Authors

---

### Author Response · Authors · 2024-11-20
**Revision**

Dear Reviewers,

We sincerely appreciate the time and effort you dedicated to reviewing our submission. Your feedback has helped to improve our paper. Below, we present the revision changes, which we marked green in the submission. Additionally, we have addressed all of your concerns raised in your reviews respectively under your reviews.

* Our paper was strongly motivated by the success of different feature engineering methods with LLMs, most notably CAAFE, which we clarified in the introduction. We additionally highlighted the influence of CAAFE in our proposed feature generation method, which we clarified further in the related work section.
* We further explained the metric “relative improvement in predictive accuracy”, which we use several times in our “Results” section when comparing different LLMs to OpenFE. We clarify that we here compare systems without feature engineering to systems with feature engineering (performed by LLMs and OpenFE, respectively).
* We reformulated the takeaways of the additional experiment. We clarified that this experiment ensures that the experienced bias towards the simple operators is not a positional bias induced by our prompt template.
* We highlight the necessity of researchers finding ways to strengthen LLMs effectively for tabular data problems so that they can be used effectively.
* We provide a critical difference plot to highlight the statistical significance of our results in Appendix G.1
* In Appendix G. 2, we compare operator selection frequencies on a subset of benchmark datasets between GPT-4o-mini and the more powerful GPT-4o to highlight that the bias is similar in these models. This leads us to believe that using more powerful models does not fix this bias.
* We provide a comparison of frequencies with which each feature of a dataset is selected by the LLMs to generate new features in Appendix G.3

---

### Author Response · Authors · 2024-12-04
**Withdrawal**

We again thank the reviewers for their time and effort to review our paper. We have considered all raised concerns in great detail and addressed each individually. Thanks to the authors' comments, we improved our manuscript in different areas, for which we are grateful.

Unfortunately, there was no possibility of discussing our changes in greater detail with the reviewers during the discussion period. Even though we addressed each reviewer's concerns individually and in great detail, we received no further comments regarding our initial responses from multiple reviewers.

We also received a high confidence rejection due to naming the experienced phenomenon we observed as “bias”. From the discussion below about this difference in opinions, we firmly believe that the word bias was justified in our case and that we provided strong background references that justify our usage of the word.
Nevertheless, we are thankful for each reviewer's comments and positive input which helped improve our work.

We have decided to withdraw our submission and are currently working on further improving this work. We aim to publish our work in the future as we firmly believe that our findings present valuable input to the community.

---

### Note · Authors · 2024-12-04

I have read and agree with the venue's withdrawal policy on behalf of myself and my co-authors.